# Trifunctional sphingomyelin derivatives enable nanoscale resolution of sphingomyelin turnover in physiological and infection processes via expansion microscopy

Marcel Rühling [1,6], Louise Kersting [2,6], Fabienne Wagner [1], Fabian Schumacher [3], Dominik Wigger[3], Dominic A. Helmerich[4], Tom Pfeuffer[2], Robin Elflein[2], Christian Kappe[5], Markus Sauer [4], Christoph Arenz [5], Burkhard Kleuser [3], Thomas Rudel [1], Martin Fraunholz [1,7] & Jürgen Seibel [2,7] ✉

Sphingomyelin is a key molecule of sphingolipid metabolism, and its enzymatic breakdown is associated with various infectious diseases. Here, we introduce trifunctional sphingomyelin derivatives that enable the visualization of sphingomyelin distribution and sphingomyelinase activity in infection processes. We demonstrate this by determining the activity of a bacterial sphingomyelinase on the plasma membrane of host cells using a combination of Förster resonance energy transfer and expansion microscopy. We further use our trifunctional sphingomyelin probes to visualize their metabolic state during infections with *Chlamydia trachomatis* and thereby show that chlamydial inclusions primarily contain the cleaved forms of the molecules. Using expansion microscopy, we observe that the proportion of metabolized molecules increases during maturation from reticulate to elementary bodies, indicating different membrane compositions between the two chlamydial developmental forms. Expansion microscopy of trifunctional sphingomyelins thus provides a powerful microscopy tool to analyze sphingomyelin metabolism in cells at nanoscale resolution.

The plasma membrane is the physical barrier defining the boundary of cells. The bilayer is comprised of various lipids, proteins, and other biological molecules[1]. Many of these components on the extracellular leaflet serve as receptors for pathogens to colonize host tissue and to invade host cells[2–6]. For the detailed understanding of infection processes of cells by viruses or bacterial pathogens fluorescence microscopy is routinely used. Whereas surface proteins can be readily fused to fluorescent proteins or are easily detected by specific antibodies, lipids

[1]Chair of Microbiology, Julius-Maximilians-University Würzburg, Würzburg, Germany. [2]Institute of Organic Chemistry, Julius-Maximilians-University Würzburg, Würzburg, Germany. [3]Institute of Pharmacy, Freie Universität Berlin, Berlin, Germany. [4]Chair of Biotechnology & Biophysics, Biocenter, Julius-Maximilians-University Würzburg, Würzburg, Germany. [5]Institute of Chemistry, Humboldt-Universität zu Berlin, Brook-Taylor-Str 2, Berlin, Germany. [6]These authors contributed equally: Marcel Rühling, Louise Kersting. [7]These authors jointly supervised this work: Martin Fraunholz, Jürgen Seibel. ✉e-mail: seibel@chemie.uni-wuerzburg.de

are rather difficult to visualize. Sphingolipids constitute one important class of bioactive plasma membrane lipids, which have been associated with the pathogenesis of Ebola virus, measles virus, SARS-CoV-2, *Pseudomonas aeruginosa*, *Staphylococcus aureus*, and many other pathogens[7–12]. The obligate intracellular pathogen *Chlamydia trachomatis*, for example, depends on the acquisition of host cell sphingolipids, especially sphingomyelin (SM), for niche formation and bacterial replication and obtains host cell SM via different transport routes[13–16].

SM is the most abundant sphingolipid in human cells. Its biosynthesis occurs in the *trans*-Golgi and the plasma membrane, which, therefore, are enriched in this lipid[17,18]. SM contains a ceramide backbone and a polar phosphocholine headgroup, that can be cleaved by sphingomyelinases (SMases). The catalytic breakdown of SM is required for membrane homeostasis and yields the hydrophobic ceramide, a central molecule of sphingolipid metabolism[19]. SMases are classified as neutral, alkaline, and acidic based on their optimum pH range. Acid sphingomyelinase (ASM) is the best-studied variant because of its link to many prevalent diseases such as major depression[20,21], cardiovascular[22], or neurodegenerative[23] diseases. Moreover, some bacterial pathogens express SMases[24,25]. These enzymes are important virulence factors, such as staphylococcal β-toxin, which is known for its lytic action towards red blood cells[26].

SMase activity can be studied by use of suitable probes. For example, assays routinely use SM in which the phosphocholine group is radiolabeled with $^{14}$C and which is either extracted with organic solvent or separated by thin-layer chromatography (TLC) followed by detection with scintillation counting[27]. Other probes carry fluorescent dyes or become fluorogenic after processing by SMase, such as BODIPY-C$_{12}$-SM or 6-hexadecanoylamino-4-methylumbelliferylphosphorylcholine (HMU-PC), respectively[28,29]. Arenz and co-workers published several Förster resonance energy transfer (FRET) probes for ASM. These probes contain two fluorescent dyes, one in the head moiety and one in the acyl chain, whereby the donor dye is excited, and by FRET the acceptor dye emits fluorescence. Upon cleavage of the probes by ASM, the fluorescence intensity of the donor increases while the FRET signal decreases. This enables ratiometric measurements indicating relative probe cleavage, independent of its concentration, which constitutes a significant advantage over traditional probes and even allows for sphingomyelinase assays in intact living cells[30–33].

Expansion microscopy (ExM) is a high-resolution microscopy technique that enables spatial resolution beyond the diffraction limit of light. Commonly, the fluorescently labeled specimen is embedded in a swellable hydrogel, which expands to multiples of its size in an excess of water. Consequently, the spatial distance between individual fluorophores is physically enlarged, increasing the resolution[34,35]. Most ExM protocols are used to image proteins, while visualization of lipids via ExM is challenging, due to their low retention within samples upon fixation and limited options for ExM-compatible staining procedures. Previously, we used a synthetic ceramide analog that enabled visualization of cellular membranes with nanoscale resolution[36].

Here, we expand the toolbox of SM probes. The derivatives that we present show only a minor deviation from the natural structure and offer three sites for further modification by click-chemistry or fixation (Fig. 1a). The phosphocholine group, as well as the ceramide part of the molecules, can be equipped with reporter molecules such as FRET pairs, allowing to determine the localization as well as the ratio of non-cleaved SM and ceramide. The third functional group can be used advantageously for ExM enabling super-resolution fluorescence imaging of the SM probe by standard fluorescence microscopy.

## Results
### Design and chemical synthesis of the SM probes
We started our investigations with the rational design of the SM derivatives. Azide and alkyne were chosen as functional groups for click-

chemistry due to their small size and their tolerance to biological systems. Hence, the derivatives should be cleavable by SMases without the impact of bulky fluorescent dyes disturbing the natural metabolism. In the first derivative (**TFSM 1**), the azide was placed in the sphingoid backbone in analogy to existing azido-sphingolipids that were successfully metabolized and studied in various biological contexts (Fig. 1a)[10–12,37–42]. In the second derivative (**TFSM 2**), the acyl chain carries the azide, while the backbone was not altered. The metabolic products of **TFSM 1** and **TFSM 2** are thus also distinguishable, which might be an advantage for mass spectrometry-based lipidomics in certain research questions. The importance of the positioning of the azide in functional sphingolipids for fluorescence microscopy was also demonstrated by Walter et al[43]. We considered that the azide in the acyl chain might be better accessible than in the backbone and aimed for characterization and comparison of both derivatives.

The second functional group for click-chemistry, the terminal alkyne, was incorporated in the phosphocholine head group based on previous findings reported in the literature. It was demonstrated by Sandbhor et al. that the introduction of an alkyne group in this position is easier than an azide due to the instability of the azido reagent. With *N,N*-dimethylpropargylamine as the nucleophile, which is a stable and commercially available substance, the terminal alkyne is easily introduced in the head group while preserving the natural quarternary ammonium group. The alkyne-modified SMs tested in the above-mentioned study were accepted as SMase substrates[44].

The additional amino function was introduced in the α-position of the acyl chain to enable ExM as demonstrated with a fixable ceramide presented by Götz et al[36].

The unique structure of the developed TFSMs allows the adjustment to several experimental setups. Compared to existing SMase probes, these molecules offer the advantage of clicking different fluorescent dyes depending on the desired application. In theory, any FRET pair can be employed if the fluorescent dyes exhibit the respective groups for click chemistry. It is important to note that the correct order of the click reactions must be respected. First, the azide in the sphingoid backbone (**TFSM 1**) or acyl chain (**TFSM 2**) can react in a strain-promoted azide-alkyne cycloaddition (SPAAC) with strained alkynes, followed by copper (I) catalyzed azide-alkyne cycloaddition of azide-containing fluorescent dyes with the alkyne-bearing phosphocholine group.

Considering the structural aspects mentioned above, we performed the synthesis of the target molecules **TFSM 1** and **TFSM 2** starting from L-serine and yielded the key intermediates **3** and **4** in multiple steps based on literature known protocols[10,45] with the stereochemistry typical for sphingolipids, consisting of an *anti*-configuration of the (protected) amino group at the *C*-2 and the alcohol at the *C*-3 position and *E* double bond geometry (Fig.1b). The secondary alcohol was protected with a methoxymethyl ether (**5** and **6**) in the subsequent step allowing an orthogonal deprotection of the primary hydroxy function (**7** and **8**). The phosphate diester bond was formed by reacting the alcohol with *β*-bromoethylphosphoryl dichloride followed by aqueous workup (**9** and **10**)[30,46]. Bromide serves as a good leaving group and was substituted by *N,N*-dimethylpropargylamine to yield the alkyne-modified phosphocholine head group (**11** and **12**)[44]. In order to introduce the acyl chain, the molecules were deprotected using HCl in 1,4-dioxane. The crude intermediates were immediately coupled with the activated pentafluorophenyl esters of the required carboxylic acids (**13** and **14**) to yield the intermediates **15** and **16**. In the last step, the Boc group was cleaved with trifluoroacetic acid, and the target compounds (**TFSM 1** and **TFSM 2**) were obtained as TFA salts.

### Metabolic acceptance of the substrate
The sterically demanding fluorophores that are pre-attached to the published visible-range FRET probe influence the acceptance of the

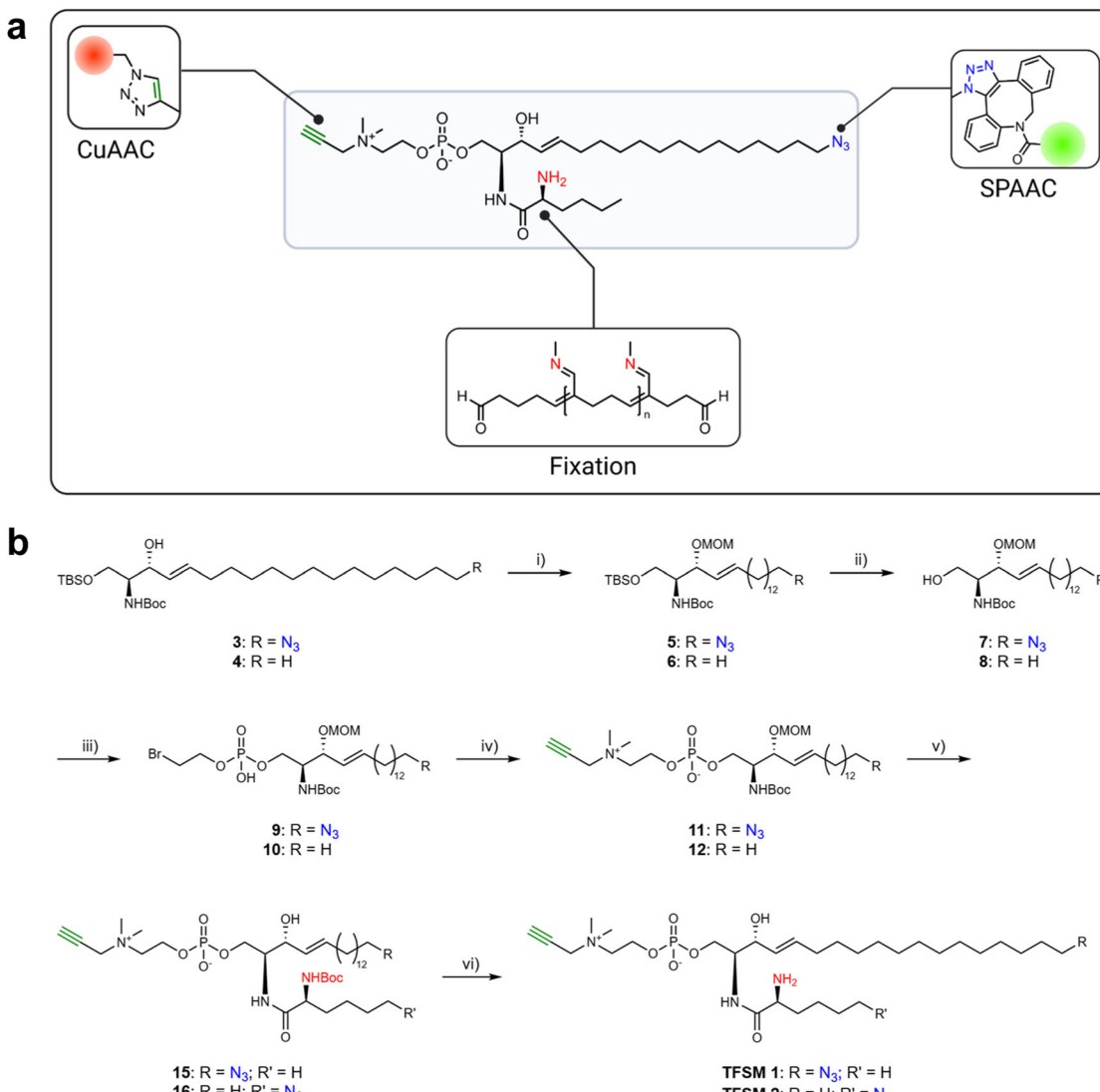

**Fig. 1 | Synthesis and applications of trifunctional sphingomyelin derivatives (TFSMs). a** Trifunctional sphingomyelin derivative **TFSM 1**. The molecule can be coupled to different conjugation partners (e.g., biotin, fluorescent dyes, quencher molecules), offering a broad range of applications. Illustrated above, a FRET pair was coupled allowing to determine the cleavage status of the probe. The third functionality is the primary amino group for fixation in expansion microscopy, in this case, glutaraldehyde fixation. **b** Synthesis of trifunctional sphingomyelin derivatives **TFSM 1** and **TFSM 2**. Synthetic route for **TFSM 1**: (i) MOM-Cl, DIPEA, DCM, 0 °C, 4 h, 85%; (ii) TBAF, THF, 0 °C, 2 h, 97%; (iii) β-bromoethylphosphoryl dichloride, pyridine, DCM, 0 °C, 4 h, 82%; (iv) N,N-dimethylpropargylamine, CHCl₃, MeCN, iPrOH, rt → 45 °C, 2 d, 63%; v) 1. HCl, MeOH, rt, 2 h; 2. **13**, DIPEA, DCM, rt, 18 h, 85% in two steps; vi) TFA, DCM, 0 °C, 2 h, quant. Synthetic route for **TFSM 2**: (i) MOM-Cl, DIPEA, DCM, 0 °C → rt, 4 h, 74%; (ii) TBAF, THF, 0 °C → rt, 1 h, 78%; (iii) β-bromoethylphosphoryl dichloride, pyridine, DCM, 0 °C, 5 h, 56%; (iv) N,N-dimethylpropargylamine, CHCl₃, MeCN, iPrOH, rt, 6 d, 69%; (v) 1. HCl, MeOH, rt, 3.5 h; 2. **14**, DIPEA, DCM, rt, 3 d, 85% in two steps; (vi) TFA, DCM, 0 °C, 2.5 h, quant. Figure 1a was created with BioRender.com, released under a Creative Commons Attribution-NonCommercial-NoDerivs 4.0 International license.

substrate by SMases other than ASM. For example, it was previously shown that the probe is not cleaved by human neutral sphingomyelinase 2 (NSM2)[31]. Similarly, in our experiments, a neutral bacterial SMase (bSMase; *S. aureus* β-toxin) did not cleave the probe (Fig. 2a). Accordingly, a probe that is specifically cleaved by neutral SMases is still missing.

The structural modifications in **TFSM 1** and **TFSM 2** are less demanding than the ones in the probe with pre-attached fluorophores. Hence, we tested whether the compounds were metabolized despite their structural modifications. We incubated HEK293T cells with 1 µM of the derivatives for 24 h and performed lipid extraction followed by LC-MS/MS analysis (Supplementary Fig. 1). The TFSM molecules were detected within the cellular lipid extracts, suggesting the uptake of the compounds by the cells (Fig. 2b). The presence of the respective ceramide metabolites

confirmed for both derivatives that they are accepted as substrates by SMases (Supplementary Figs. 2, 3). For **TFSM 1** we also measured the sphingosine metabolite indicating processing by ceramidases. Since metabolization of **TFSM 2** by ceramidases yields canonical sphingosine, it remains elusive whether the compound is accepted by ceramidases. The sphingosine content shown for **TFSM 2** (Fig. 2b) thus represents the sum of intrinsic sphingosine and sphingosine supposedly originating from **TFSM 2**. For cells treated with **TFSM 2**, we found a slight increase in sphingosine levels compared to the solvent control (normalized peak area: 0.652 vs 0.489), although this was not statistically significant ($p = 0.2687$).

In addition, we detected higher amounts of the **TFSM 2** ceramide metabolite than for **TFSM 1**, suggesting that **TFSM 2** is metabolized more efficiently by SMases (Fig. 2c).

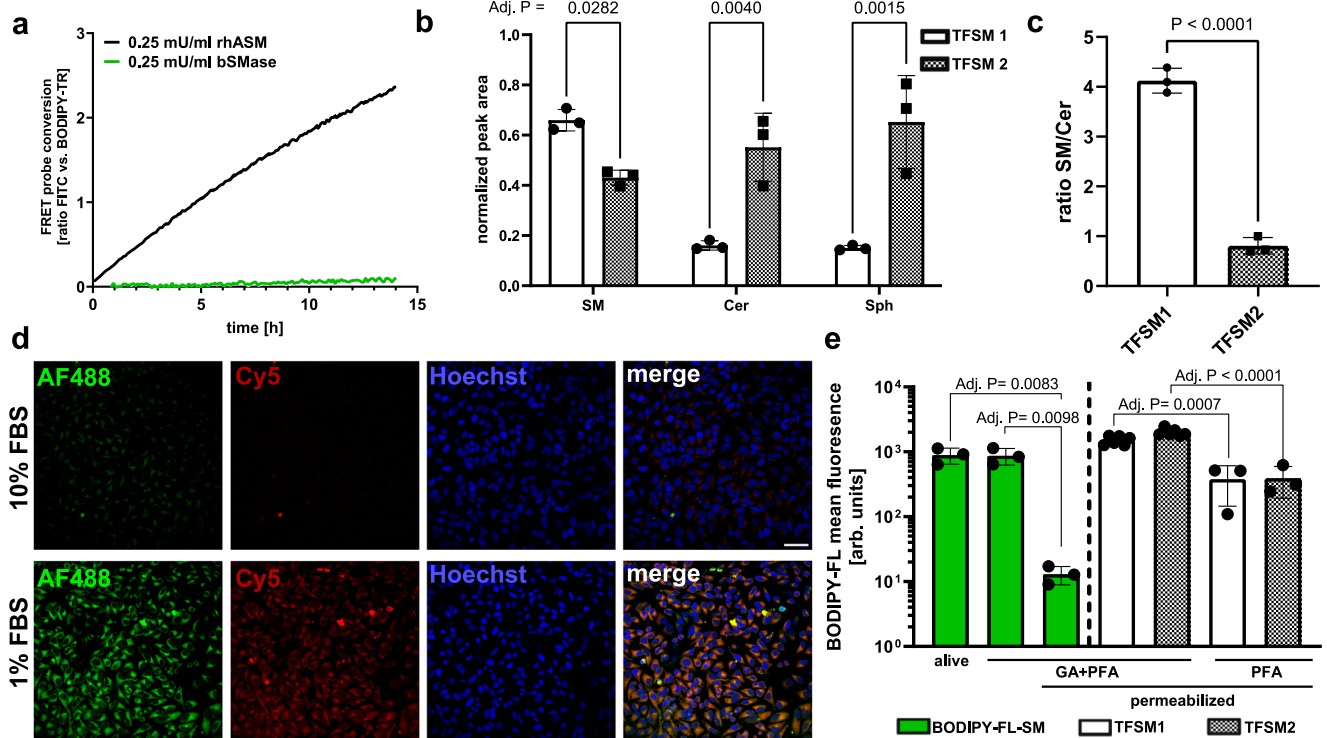

**Fig. 2 | Trifunctional sphingomyelins (TFSMs) are taken up and metabolized by human cells. a** Sphingomyelinase Förster resonance energy transfer (FRET) probe is cleaved by recombinant human acid sphingomyelinase (rhASM) but not bacterial sphingomyelinase (bSMase). rhASM or bSMase were incubated in the presence of FRET probe and probe conversion was monitored by measurement of FITC fluorescence in a microplate reader. $n = 3$. **b, c** TFSMs are incorporated into human cells. HEK293T cells were treated with **TFSM 1** or **TFSM 2** for 24 h at 37 °C. After cell lysis, amounts of TFSMs, ceramide (Cer), and sphingosine (Sph) were detected by LC-MS/MS. Internal standards were used for each metabolite type, and their peak areas were used to normalize the peak area of the corresponding TFSM metabolite. $n = 3$. **d** Cellular uptake of TFSMs is affected by fetal bovine serum (FBS). HeLa cells were treated with **TFSM 1** in the presence of 1 or 10% FBS for 2 h. Samples were fixed and clicked with AlexaFluor™ (AF)488-DBCO (backbone) or Cy5-azide (headgroup). $n = 1$. Scale bars: 50 µm. **e** Glutaraldehyde (GA) fixation of TFSMs. HeLa cells were incubated with BODIPY-FL-$C_{12}$-sphingomyelin (BODIPY-FL-$C_{12}$-SM), **TFSM 1** or **TFSM 2** for 2 h, bSMase-treated, detached, fixed, and permeabilized with TX-100 (as indicated). Fixation was done either with paraformaldehyde (PFA) or PFA and GA. TFSM then were BODIPY-FL-DBCO-stained and analyzed by flow cytometry. $n = 3$ (BODIPY-FL-$C_{12}$-SM, **TFSM 1** PFA and **TFSM 2** PFA), $n = 6$ (**TFSM 1** GA + PFA, **TFSM 2** GA + PFA). Statistics: Two-way ANOVA and Šídák's (**b**) or Tukey's (**e**) multiple comparisons. Two-sided unpaired Student's $t$ test (**c**). Bars represent means ± SD. n corresponds to biological replicates. Source data and detailed statistics are provided as a Source Data file.

## Visualization of cellular membranes

Next, we established a protocol for staining cellular membranes with TFSMs. First, we tested whether the cellular uptake of the molecules is sufficient for detection in fluorescence microscopy. Therefore, we treated HeLa cells with 10 µM **TFSM 1** in the presence of 10% fetal bovine serum (FBS), which is a concentration that is commonly used in cell culture. The cells were fixed, permeabilized, and clicked with AlexaFluor™ 488-DBCO as well as Cy5 azide in the presence of Cu(I). Thereby, we only detected a weak fluorescence signal (Fig. 2d; 10% FBS). Since sphingolipids can be bound by bovine serum albumin (BSA), a major component of FBS[47,48], we first investigated how the presence of FBS affects the cellular uptake of the previously published visible range FRET probe[31]. We treated HeLa cells in the presence of varying FBS concentrations and measured BODIPY-TR fluorescence, which correlates to probe uptake, via microscopy (Supplementary Fig. 4a) or flow cytometry (Supplementary Fig. 4b). We found that FBS reduces probe uptake in a concentration-dependent manner. Next, we treated HeLa cells with **TFSM 1** in the presence of 1% FBS, fixed and permeabilized, and clicked with AlexaFluor™488-DBCO and Cy5 azide. The staining of cellular membranes by the molecule (Fig. 2d; 1% FBS) was enhanced compared to a sample that was treated in the presence of a higher FBS concentration (Fig. 2d; 10% FBS).

We also investigated the retention of TFSMs and BODIPY-FL-$C_{12}$-SM within samples upon permeabilization. Therefore, we incubated HeLa cells with BODIPY-FL-$C_{12}$-SM and either analyzed the cells alive, fixed, or fixed + permeabilized for BODIPY-FL fluorescence via flow cytometry (Fig. 2e). While retention of the lipid was not affected by fixation, permeabilization drastically reduced BODIPY-FL fluorescence. Next, we incubated HeLa cells with **TFSM 1** and **TFSM 2**, fixed either with glutaraldehyde (GA) + paraformaldehyde (PFA) or only with PFA, permeabilized, clicked with BODIPY-FL-DBCO and analyzed by flow cytometry (Fig 2e) or microscopy (Supplementary Fig. 4c). We measured high retention of both compounds when we fixed with GA + PFA, while BODIPY-FL fluorescence was markedly reduced in samples solely fixed with PFA as already observed by a previous study[36].

We observed enrichment of **TFSM 1** and **TFSM 2** fluorescence at distinct sites within the cells. **TFSM 1**-enriched sites thereby colocalized with the Golgi matrix protein (GM) 130 (Fig. 3a), suggesting a high abundance of the compounds within the Golgi apparatus. We next tested, whether TFSMs would colocalize with other membrane-rich organelles and stained the mitochondrial matrix protein Peroxiredoxin 3 (Prx3). We observed colocalization of **TFSM 1** and **TFSM 2** with mitochondria in HeLa and primary human umbilical vein endothelial cells (HuVEC; Fig. 3b and Supplementary Fig. 5).

## Measuring the activity of a neutral bSMase on human cells

Probes for the detection of ASM activity are available[30,31], whereas a probe that enables measurement of the activity of a SMase on the

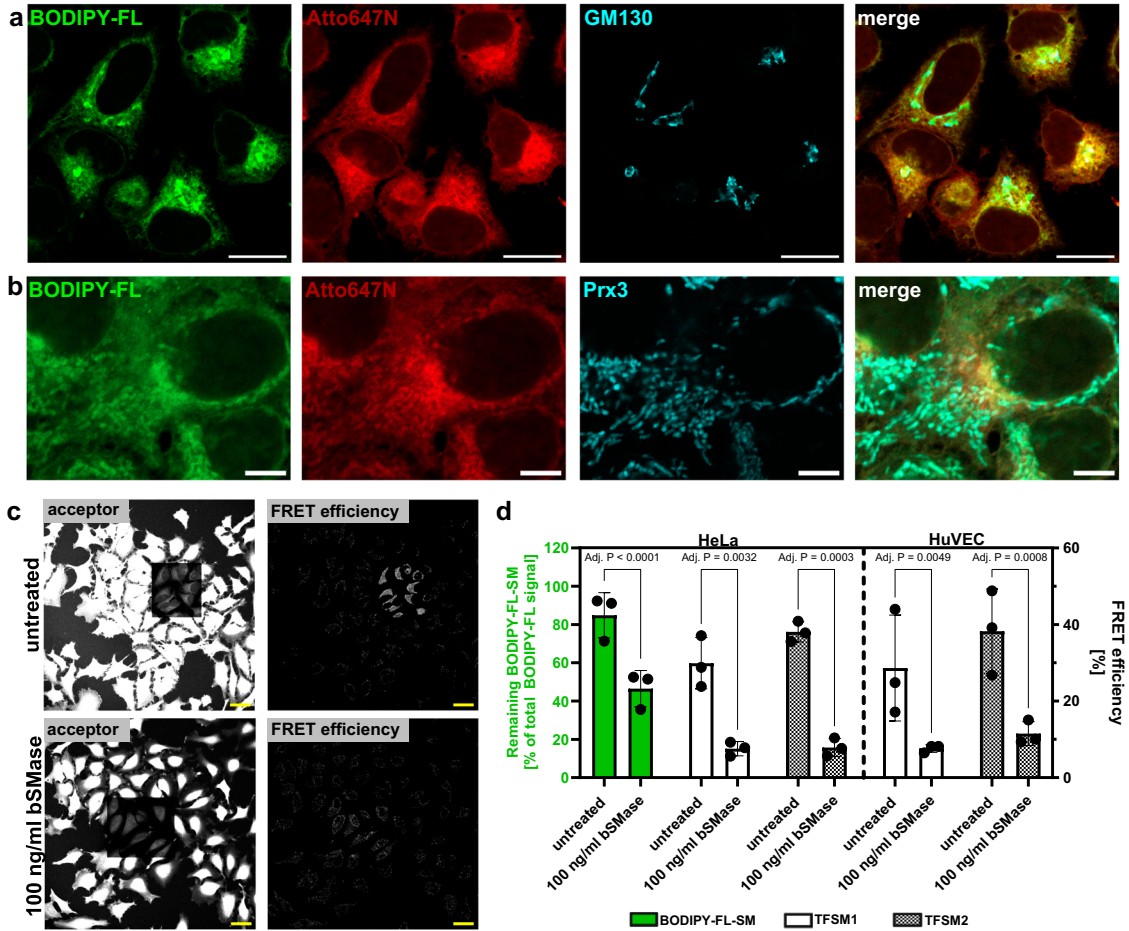

**Fig. 3 | Trifunctional sphingomyelins (TFSMs) are incooperated in cellular compartments and can be used to determined bacterial sphingomyelinase (bSMase) activity. a, b** TFSMs are enriched within the Golgi and mitochondria. HeLa cells were treated with **TFSM 1** for 2 h (**a**) or 24 h (**b**). Samples were fixed and clicked to BODIPY-FL-DBCO (backbone) or Atto647N-azide (headgroup). Golgi and mitochondria were visualized by an anti-Golgi matrix protein 130 (GM130) and anti-Peroxiredoxin (Prx3) antibody, respectively. $n = 1$. Scale bars: 25 µm (**a**) or 5 µm (**b**). **c, d** FRET measurement of TFSM conversion. HeLa or human umbilical vein endothelial cells (HuVEC) were incubated with **TFSM 1, TFSM 2,** or BODIPY-FL-$C_{12}$-

SM for 2 h. Then, the compounds were removed, and cells were either treated with bSMase or left untreated for 3 h. Samples treated with TFSMs were fixed, clicked with BODIPY-FL-DBCO (backbone) and AlexaFluor™546 (AF)546-azide (headgroup) and FRET efficiency was determined by acceptor bleaching. Cells treated with BODIPY-FL-$C_{12}$-SM were detached, and lipids were extracted by $CHCl_3$:MeOH. The proportion of unmetabolized SM was determined by thin-layer chromatography. Scale bars: 25 µm. $n = 3$. Statistics: Two-way ANOVA and Šídák's multiple comparisons (**d**). Bars represent means ± SD. n corresponds to biological replicates. Source data and detailed statistics are provided as a Source Data file.

plasma membrane is missing. To test whether we could visualize cleavage of the TFSMs by the staphylococcal neutral bSMase β-toxin, we incubated HeLa with **TFSM 2** for 2 h, removed the molecule, and treated the cells with bSMase for 3 h. The samples were fixed, and BODIPY-FL-DBCO was clicked to the backbone while AlexaFluor™546-azide was clicked to the headgroup of the molecule. We next conducted FRET acceptor bleaching in selected regions of interest (ROI; Fig. 3c, see Supplementary Fig. 4d for **TFSM 1**) and measured donor fluorescence pre- and post-bleaching in the respective area to determine the FRET efficiency. For both compounds, we determined a FRET efficiency of ~30%, which was markedly reduced upon treatment with bSMase (Fig. 3d). Thus, we concluded that the bSMase cleaved off the modified phosphocholine headgroup, thereby abolishing FRET with the fluorescence acceptor in the TFSMs. We compared our results with a conventional method, where we incubated HeLa with BODIPY-FL-$C_{12}$-SM, extracted the lipid, and determined the proportion of remaining unmetabolized BODIPY-FL-$C_{12}$-SM via thin-layer chromatography (Fig 3d and Supplementary Fig. 4e). The amount of SM was reduced upon bSMase treatment in the same fashion as observed for TFSMs.

## Visualizing the cleavage of TFSMs with subcellular resolution by 4xExM

Next, we monitored TFSM turnover by ExM. Therefore, HeLa cells were incubated for 24 h with 10 µM **TFSM 1** in the presence of 1% FBS. Samples were fixed, clicked with BODIPY-FL-DBCO as well as AlexaFluor™546-azide, and were 4x expanded. Confocal fluorescence microscopy demonstrated the staining of cellular membranes. Clearly, 4-fold expansion enabled the visualization of cellular membranes at higher spatial resolution (Fig. 4a), thus enabling imaging of the mitochondrial envelope (Supplementary Fig. 6).

Acceptor photobleaching of the samples demonstrated an increase of donor fluorescence intensity in the bleached ROIs (Fig. 4b), resulting in a FRET efficiency of ~33%, which decreased upon treatment with bSMase (Fig. 4c). Thus, TFSM can be used to monitor cleavage by SMases.

We observed spots with enhanced FRET signals in our samples, suggesting that these sites contain a high proportion of non-processed TFSM. Since lysosomes are the major site of SM degradation, we immuno-stained the sample for lysosomal-associated membrane protein 1 (LAMP1) and observed colocalization

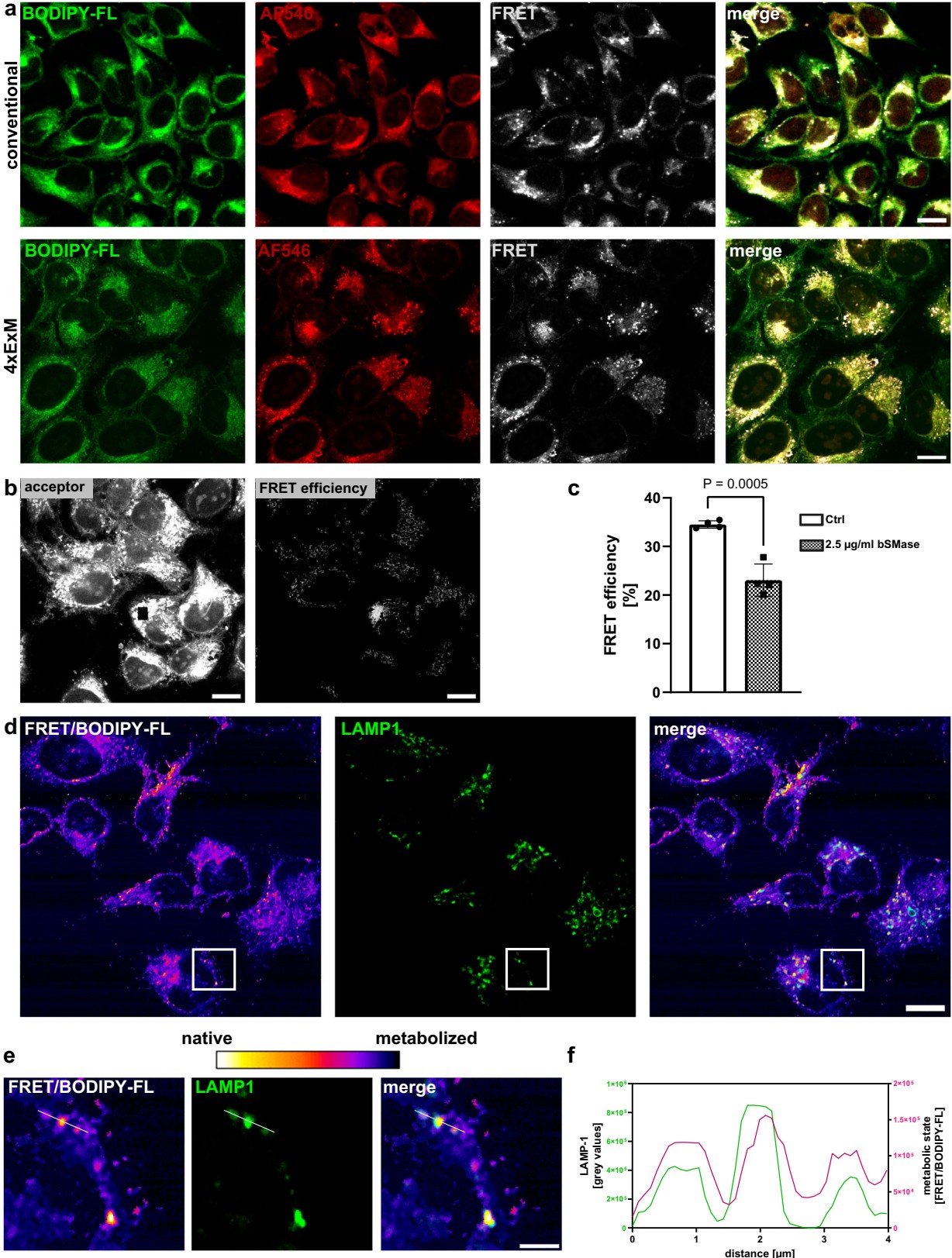

(Fig. 4e, f). Moreover, we calculated ratio images by dividing the FRET and donor channels to estimate the metabolic state of the TFSM derivative with high spatial resolution (Fig. 4d). Together, our results demonstrate the presence of non-metabolized TFSM in lysosomes (Fig. 4e, f).

## TFSMs enable visualization of sphingolipid metabolization during chlamydial infection

The obligate intracellular pathogen *C. trachomatis* forms large inclusions within its host cells and, thereby, is highly dependent on host metabolites, such as sphingolipids. In a previous study, we showed that

**Fig. 4 | Trifunction sphingomyelins (TFSMs) can be used for expansion microscopy (ExM). a** TFSMs are compatible with ExM. HeLa cells were incubated with **TFSM 1** for 24 h in the presence of 1 % FBS. Samples were fixed, stained with BODIPY-FL-DBCO as well AlexaFluor™ (AF)546-azide, and then either imaged by conventional CLSM or 4-fold expanded. $n = 3$. Scale bars: 50 μm (~12.5 μm with 4-fold expansion factor). **b, c** Förster resonance energy transfer (FRET) can be measured in expanded samples. HeLa cells were incubated with **TFSM 1** for 24 h in the presence of 1% FBS. Then, the compound was removed, and cells were treated with 2.5 μg/ml bacterial sphingomyelinase (bSMase) or left untreated for 24 h. Samples were fixed, stained, and expanded as described in (**a**) and analyzed via acceptor photobleaching to determine FRET efficiency. $n = 4$. **d–f** Lysosomes possess high SM content. HeLa cells were treated as described in (**a**), and lysosomal-associated membrane protein 1 (LAMP1) was stained by an anti-LAMP1 primary and a CF568 secondary antibody. Images recorded in the FRET channel were multiplied by factor 2 and divided by the BODIPY-FL (donor) channel resulting in images that describe the metabolic state of the molecule. $n = 1$. Scale bars: 50 μm (~12.5 μm with 4-fold expansion factor) and 12 μm for zoomed images (~3 μm with 4-fold expansion factor). Statistics: Two-sided unpaired Student's $t$ test (**c**). Bars represent means ± SD. n corresponds to biological replicates. Source data and detailed statistics are provided as a Source Data file.

a fixable and clickable $C_6$-ceramide is taken up by the bacteria[36]. Furthermore, uptake of SM derived from $C_6$-NBD-ceramide into the chlamydial inclusion was shown[15]. To further evaluate sphingolipid metabolism and uptake into chlamydial inclusions, we infected HeLa cells with *C. trachomatis* at MOI 1. To avoid direct contact of bacteria with TFSMs, the medium was changed 3 h post-infection (p.i.), and the TFSMs were added subsequently. At 27 h p.i., cells were fixed, stained with BODIPY-FL-DBCO and AlexaFluor™546-azide, and analyzed via conventional CLSM. We observed a high BODIPY-FL signal within inclusions (Fig. 5a; white lines), whereas only a moderate FRET signal was detected compared to host cell areas. To visualize the metabolic state of the molecule, we divided the FRET by the donor (BODIPY-FL) channel. We found an increased proportion of metabolized molecules in chlamydial inclusions compared to host cell areas. Next, we measured the FRET efficiency either in an area containing host cell membranes or a chlamydial inclusion (Fig. 5b). A FRET signal could be detected within the inclusion, suggesting the presence of the non-metabolized TFSMs. However, the FRET efficiency was significantly higher in areas containing host cell membranes, indicating that inclusions possess a higher proportion of TFSM metabolites (Fig. 5c).

To further confirm our observations, we conducted fluorescence lifetime imaging microscopy (FLIM) with *Chlamydia*-infected samples. We observed a reduced fluorescence lifetime of the donor fluorophore BODIPY-FL in the presence of the acceptor clicked to **TFSM 1** (Fig. 5d, e, Supplementary Fig. 7 and Supplementary Table 1), indicating FRET between the fluorophores. Furthermore, we detected higher fluorescence lifetimes in chlamydial inclusions compared to regions containing host cell membranes (Fig. 5d), again suggesting a high proportion of metabolized TFSM within inclusions.

Inclusions harbor two developmental forms of *C. trachomatis*: the larger metabolically active reticulate bodies (RBs) as well as the smaller infectious elementary bodies (EBs)[49]. Despite intense research, the membrane composition of RBs and EBs remains elusive. To elucidate the distribution of the molecule within an inclusion, we conducted 4xExM with a *Chlamydia*-infected sample and stained **TFSM 1** with BODIPY-FL-DBCO and AlexaFluor™546-azide. Moreover, we stained chlamydial HSP60 to visualize the inclusion (Fig. 6a). As previously observed with unexpanded samples, the inclusion possesses a lower proportion of native molecule, which can be visualized by the metabolic state (ratio FRET vs. donor/BODIPY-FL channel, Fig. 6b and Supplementary Movie 1). This was further confirmed by acceptor photobleaching of chlamydial inclusions or regions containing host cell membranes (Fig. 6c). Thereby, we observed a lower FRET efficiency within the inclusion area, suggesting the presence of a high proportion of the metabolized molecule (Fig. 6d).

Nevertheless, we were able to detect a FRET signal within the inclusion membranes as well as RBs that localize to the periphery of the inclusion. Although the FRET signal in RBs was lower compared to host cell membranes, it was clearly higher than the unspecific background signal. Hence, we concluded that TFSM is also present in the non-metabolized state within *C. trachomatis*, albeit to a lower extend, as in host cell membranes. In contrast to RBs [Fig. 6e, f; i.)], the smaller, less metabolic-active EBs, which redifferentiate from RBs during infection, predominantly possessed a higher proportion of donor

signal, whereas the FRET signal was comparable [Fig. 6e, f; ii.)] or even lower [iii.)] than in RBs. This indicates a particularly high content of metabolized **TFSM 1** within EBs. We also rarely found particles that featured characteristics of RBs and EBs (Fig. 6b, white arrows). These bacteria owned a comparably high proportion of non-metabolized TFSMs as found in RBs, while their size rather resembled EBs. Similarly, we detected a higher proportion of metabolized **TFSM 2** within EBs (Supplementary Fig. 8).

Next, we conducted FLIM of 4-fold expanded samples that were infected with *Chlamydia*. Thereby, we observed reduced fluorescence lifetimes in areas containing host cell membranes, whereas lifetimes within chlamydial inclusions were longer, even though the differences were less pronounced as previously seen in unexpanded samples (Supplementary Fig. 7 and Supplementary Table 1). Again, the decreased FRET signal indicates a lower proportion of native non-metabolized **TFSM 1** within the inclusions. The highest fluorescence lifetimes within inclusions were measured in EBs (Fig. 5g and Supplementary Fig. 9), again indicating that EBs predominantly contain metabolized TFSM.

## Membrane-integral fluorophores are affected by their lipid environment

Next, we tested if metabolization of TFSMs can be detected by ratiometrically analyzing two fluorophores without using a FRET system. Therefore, HeLa cells were incubated with **TFSM 1**, treated with bSMase, and stained with AlexaFluor™488-DBCO and Cy5-azide, two fluorophores that do not interact via FRET (Fig. 7a). As expected, the fluorescence intensity of Cy5, clicked to the modified phosphocholine head group, was reduced upon bSMase treatment. For quantification, we measured the fluorescence intensity of both Cy5 as well as AlexaFluor™488 channels and calculated the ratio (Cy5 vs. AlexaFluor™488). In principle, the obtained ratios describe the metabolic conversion of the TFSMs, whereby a high ratio indicates a large proportion of the original non-metabolized **TFSM 1** (Fig. 7b). As expected, upon bSMase treatment, the ratio was decreased compared to untreated controls, suggesting enzymatic turnover. Hence, in principle, bSMase treatment can be monitored ratiometrically.

However, we surprisingly observed an enhanced AlexaFluor™488 signal in samples treated with bSMase (Fig. 7a), even though the presence of Cy5 should not affect AlexaFluor™488 fluorescence.

To further investigate this phenomenon, we established a flow cytometry-based readout for TFSMs. We incubated HeLa with **TFSM 1** for 2 h, removed the molecule again, and treated the cells with 100 ng/ml bSMase or left them untreated. Then, we detached the cells and stained the backbone of **TFSM 1** with BODIPY-FL DBCO, while we did not click another dye to the headgroup. Subsequently, BODIPY-FL mean fluorescence was analyzed via flow cytometry (Fig. 7c). Again, bSMase-treated cells possessed higher BODIPY-FL signals. Since there was no fluorophore clicked to the headgroup of **TFSM 1**, the observed change in fluorescence intensity cannot be due to intramolecular FRET.

Next, we tested whether this phenomenon can be observed with other SM derivatives and repeated the experiment with the visible-range FRET probe with pre-attached fluorophores. We detected an

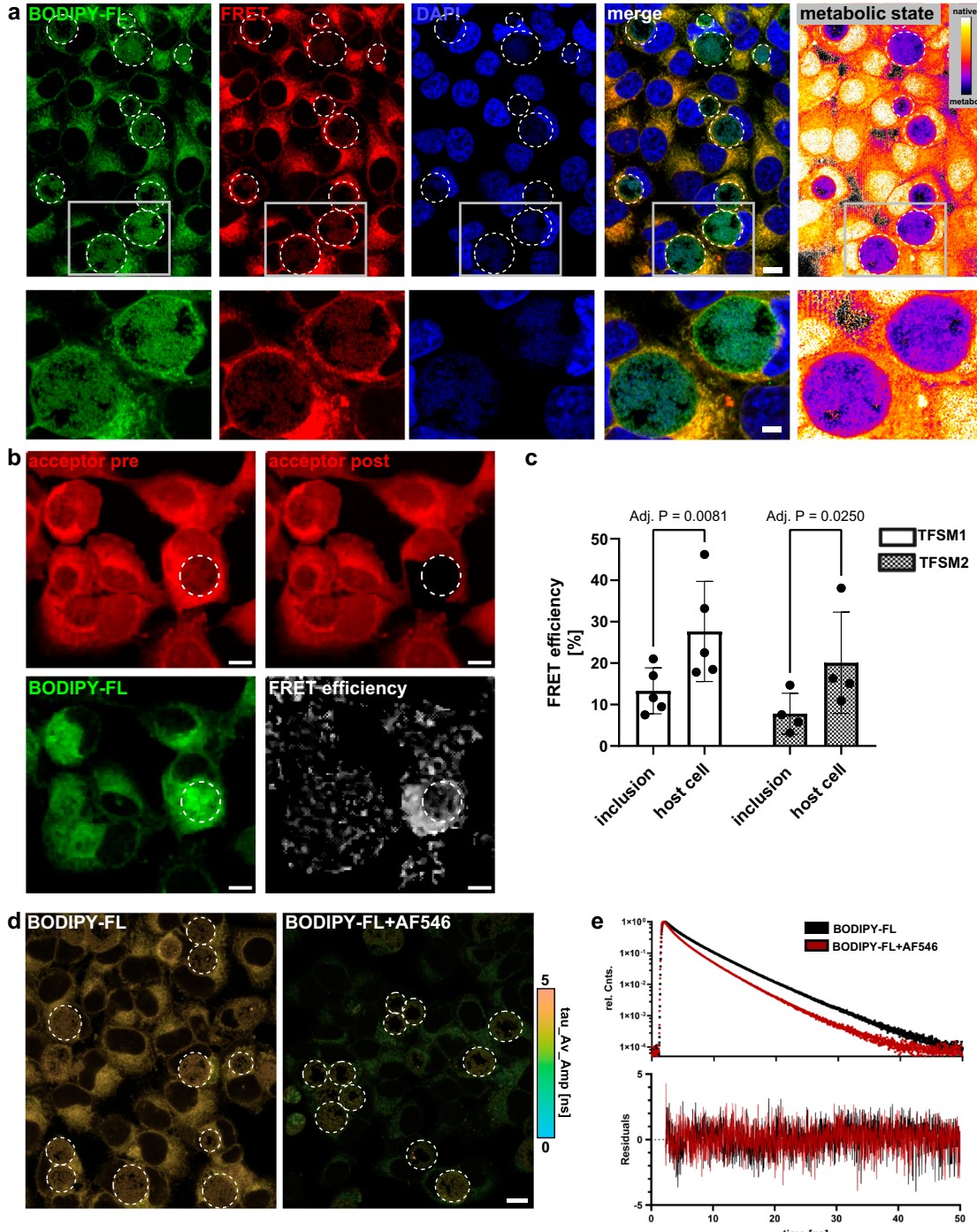

**Fig. 5 | *Chlamydia* inclusions are enriched with the metabolized trifunctional sphingomyelins (TFSMs). a** *Chlamydia* inclusions possess a higher proportion of metabolized **TFSM 1** than host cell membranes. HeLa cells were infected with *C. trachomatis* at a multiplicity of infection (MOI) 1 in the presence of 10 μM **TFSM 1** and 1% FBS. After 24 h of treatment, cells were fixed, stained with BODIPY-FL-DBCO (backbone) and AlexaFluor™ (AF)546-azide (headgroup), and samples were imaged with a confocal microscope. The metabolic state of TFSM is described by the ratio of 3x Förster resonance energy transfer (FRET) channel vs. donor channel (BODIPY-FL). Chlamydial inclusions are indicated with white circles. *n* = 5. Scale bars: 10 μm, zoomed: 5 μm. **b**, **c** Chlamydial inclusions possess lower FRET efficiency than host cell membranes. Samples were prepared as described in (**a**), and FRET efficiency was determined by acceptor bleaching. The acceptor fluorophore (AF546) was bleached in an area either containing a chlamydial inclusion (white

line) or host cell membranes. Then, donor fluorescence before (pre) and after (post) bleaching was compared to determine FRET efficiency. In (**b**), samples incubated with **TFSM 2** are depicted. Scale bars: 10 μm. **TFSM 1:** *n* = 5, **TFSM 2** *n* = 4. **d** Overview fluorescence lifetime imaging microscopy (FLIM) images of infected, **TFSM 1**-stained HeLa cells click-labeled with BODIPY-FL-DBCO (left) as well as BODIPY-FL-DBCO and AF546-azide (right) measured by confocal imaging at an irradiation intensity of 0.5 kW cm⁻². *Chlamydia* inclusions are marked with white lines. No intensity threshold was applied. *n* = 1. Scale bars: 20 μm. (**e**) Average fluorescence decays from 9 individual FLIM images of single-cell measurements shown in Supplementary Fig. 7. *n* = 1. Statistics: Mixed effects analysis (REML) and Šídák's multiple comparisons test (**c**). Bars represent means ± SD. n corresponds to biological replicates. Source data and detailed statistics are provided as a Source Data file.

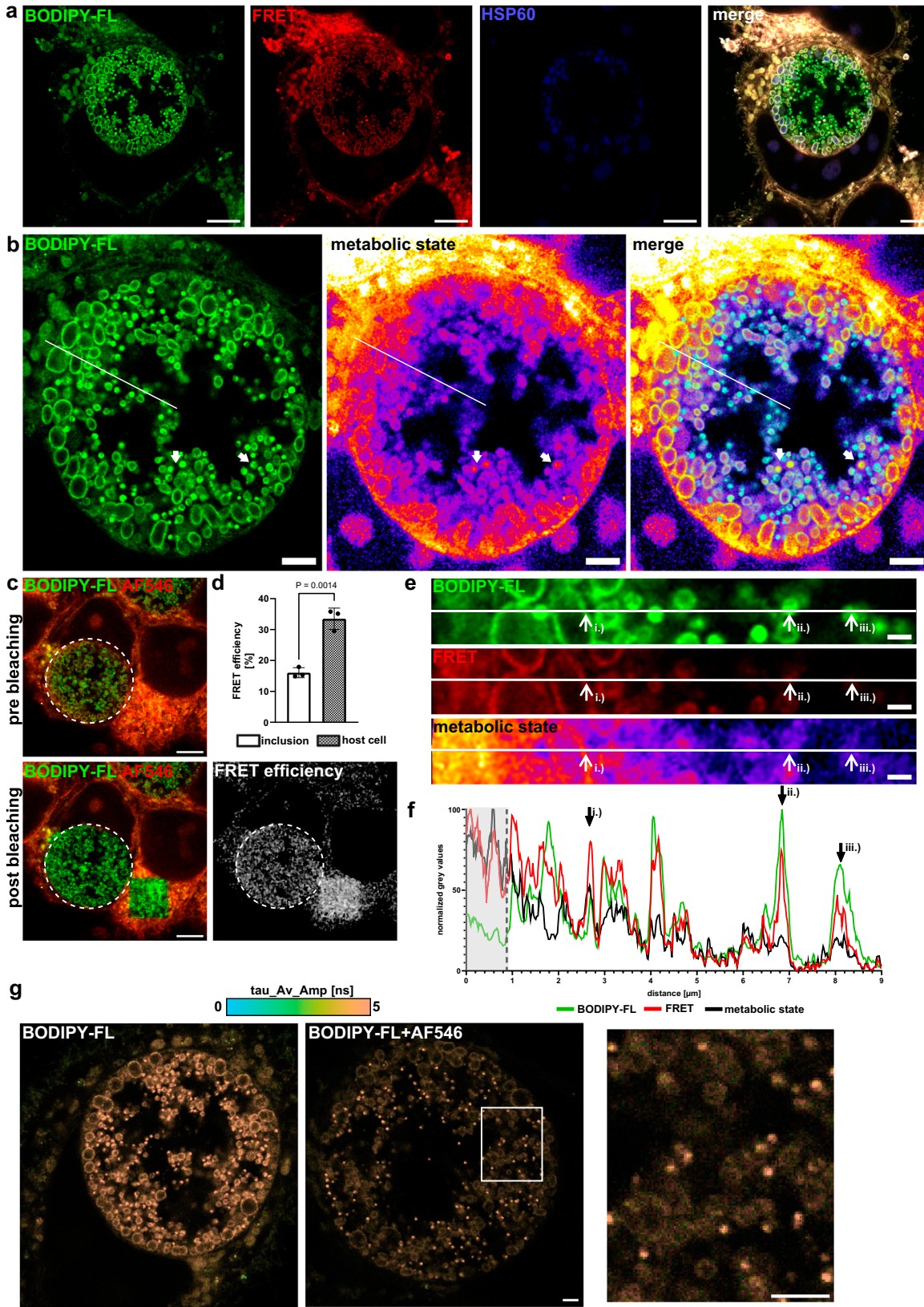

increased BODIPY-TR fluorescence upon bSMase treatment, whereas the FRET system was not affected, thereby confirming that the FRET probe is not metabolized by the bSMase (Fig. 7d). A similar but less pronounced increase of fluorescence was also observed, when we conducted the experiments with a commercially available BODIPY-FL-C$_{12}$-SM analog (Fig. 7e). Thus, we concluded that the enhanced signal

of membrane-associated fluorophores of our SM derivatives does not necessarily require metabolization of the molecules themselves.

Since close proximity of fluorophores can result in self-quenching, we performed ExM on our samples, which physically enlarged the spatial distance between molecules within the specimen. Hence, self-quenching should be reduced in expanded samples.

**Fig. 6 | Elementary bodies possess higher proportion of metabolized TFSMs than reticulate bodies. a** TFSM can be used to visualize chlamydial inclusion via 4xExM. HeLa cells were infected with *C. trachomatis* at MOI 1 in the presence of **TFSM 1** for 24 h, fixed and stained with AlexaFluor™ (AF)546-azide (headgroup) and BODIPY-FL-DBCO (backbone). *Chlamydia* was stained with an anti-chlamydial heat-shock protein 60 (HSP60) anti-body and an AF405 secondary antibody. Samples were 4x expanded and imaged with CLSM. $n = 3$. Scale bars: 8 μm (with 4-fold expansion factor ~ 2 μm). **b** Periphery of inclusions rather possess non-metabolized TFSM compared to their center. The image depicted in (**a**) was zoomed, and the metabolic state of TFSM was calculated by determining the ratio of 3-times Förster resonance energy transfer (FRET) vs. donor (BODIPY-FL) signal. White arrows indicate intermediate chlamydial developmental forms. $n = 3$. Scale bars: 8 μm (with 4-fold expansion factor ~ 2 μm). **c, d** Inclusions possess lower FRET efficiency than host cell membranes. *Chlamydia*-infected HeLa cells stained with **TFSM 1** (BODIPY-FL: Donor/AF546: Acceptor) were 4-fold expanded. FRET efficiency in regions containing an inclusion (white circles) or host cell membranes was determined by acceptor photobleaching. $n = 3$. Scale bars: 20 μm (with 4-fold expansion factor ~ 5 μm). **e, f** Sphingolipid composition of EBs and RBs differ. The area indicated by a white line in b.) was magnified, and the intensity profile was measured in Fiji. The resulting gray values were scaled to the highest and lowest values detected in individual channels. $n = 3$. Scale bar: 4 μm (with 4-fold expansion factor ~ 1 μm). **g** Identification of high metabolized TFSM levels in EBs by FLIM imaging of 4-fold expanded **TFSM 1**-stained chlamydial inclusion labeled with BODIPY-FL-DBCO (left) as well as BODIPY-FL-DBCO and AF546-azide (middle, right). Zoomed image (right) of BODIPY-FL-DBCO and AF546-azide labeled TFSM shows higher fluorescence lifetimes of EBs in comparison to RBs, indicating a higher proportion of metabolized TFSMs. $n = 1$. Scale bars: 10 μm (with 4x expansion factor ~ 2.5 μm). Statistics: Two-sided unpaired Student's *t* test (**d**). Bars represent means ± SD. n corresponds to biological replicates. Source data and detailed statistics are provided as a Source Data file.

We therefore incubated HeLa cells with **TFSM 1** for 24 h, removed the molecule, and treated the cells with bSMase. Subsequently, the backbone was clicked with BODIPY-FL-DBCO, and samples were either embedded in Mowiol or a hydrogel. The gelated samples were either left unexpanded or were expanded. We observed an increase in BODIPY-FL fluorescence upon bSMase treatment in images obtained in Mowiol and Gel-embedded unexpanded samples (Fig. 6f). In contrast, there were no differences between bSMase-treated samples and untreated controls after 4-fold expansion (Fig. 6g). We concluded that the enhanced signal of membrane-integral fluorophores upon bSMase treatment is due to a larger intermolecular distance between membrane-integral fluorophores.

## Discussion

Here we present the synthesis and the thorough characterization of TFSM derivatives that possess two functional groups for click-reactions and an amino group for fixation and demonstrate their versatile applications. The TFSMs are taken up and metabolized by human cells as we confirmed by LC-MS/MS analysis. We included a FRET probe[31] in our investigations and showed that the TFSMs complement this probe in terms of SMase specificity. Due to the comparably small chemical modifications in TFSMs, they can be metabolized by the neutral bSMase β-toxin, whereas the FRET probe with bulky pre-attached fluorophores was not accepted by this enzyme. Nevertheless, the FRET probe benefits from high specificity for ASM and, hence, enables measurement of cellular ASM activity in the presence of other SMases such as NSM2. Moreover, the pre-attached fluorophores enable monitoring of ASM activity in living cells without the need for fixation and staining steps, which can be cytotoxic such as copper-catalyzed click reactions that are required for visualization of TFSMs. However, unlike TFSMs, the FRET probe does not possess a primary amine function, thus limiting its applicability for fluorescence microscopic applications.

We show that BODIPY-FL-C$_{12}$-SM, a lipid not possessing a primary amine function, was not retained within the sample upon permeabilization, a step that is required for standard expansion protocols. By contrast, TFSMs were retained in specimens upon permeabilization, if we fixated the samples with GA and PFA, demonstrating the importance of the primary amine group as observed previously[36]. Hence, our approach is compatible with standard ExM protocols and needs no further adjustment. However, there are other protocols that enable ExM of lipids involving either the usage of specialized linker molecules[50,51], streptavidin labeling[52], or methacrolein fixation[53], which serve as alternatives to equipping the lipid analogs with amine functions.

By FRET acceptor bleaching and FLIM, we clearly showed that the two fluorophores that were attached to the TFSMs via click-chemistry interact via FRET and that the FRET system is destroyed by the bSMase, strongly suggesting that the enzyme metabolizes TFSMs. In theory, FRET is not compatible with ExM, for instance, to study protein-protein interaction, since fluorophores that are attached to two individual molecules are teared from each other during sample expansion, thereby preventing FRET. However, in our approach presented in this study, both fluorophores are attached to the same molecule, enabling the detection of intramolecular FRET in expanded samples, as we show by acceptor photobleaching and FLIM.

The combination of TFSMs with CLSM allowed the detection of the compounds' metabolic states with nanoscale spatial resolution. For instance, we detected a high proportion of native TFSMs in lysosomes, while chlamydial inclusions rather possess the metabolized compound.

Previous studies have shown that *C. trachomatis* incorporates exogenously added sphingolipids into its inclusion[36,54]. Hackstadt and colleagues demonstrated the uptake of NBD-C$_6$-ceramide into the inclusion, as well as endogenously synthesized NBD-C$_6$-SM[15]. Various studies presented the importance of SM for bacterial replication and inclusion stability[13,14,55]. SM can be acquired from the host via different non-vesicular and vesicular routes[56], like the transport of ceramide via the so-called ceramide transfer protein (CERT) from the ER to the chlamydia inclusion[16]. The recruitment of host SM synthases[56] or an endogenous chlamydial SM synthase is required for SM generation from ceramide[57]. Although the lipid composition of chlamydial inclusions was intensively studied, less is known about the membrane constitution of RBs and EBs. In the late 90 s, Wylie et al. reported a similar lipid profile for both developmental forms, except for cardio-lipin, which was found enriched in EBs[54]. However, the separation of EBs and RBs, which was central to this study for determining lipid quantities in either developmental form, is challenging, and the observations might be biased by cross-contaminations. Moreover, the study did not investigate chlamydial ceramide content.

By ExM, we herein visualized the spatial distribution of native and metabolized TFSM within a chlamydial inclusion. We observed a higher proportion of native TFSM in the periphery of the inclusion, whereas in the center metabolized TFSMs were enriched. In particular, we found TFSM metabolites highly enriched in EBs, while cell membranes of RBs possessed larger amounts of the native compound. Since EBs develop from RBs, we hypothesized that TFSMs are cleaved by an SMase during EB maturation. In consistence, a previous study reported SMase activity within EBs of *Chlamydia pneumonia*, suggesting the expression of a chlamydial SMase[58]. Moreover, we observed intermediate forms that possess the size of EBs but still contain native TFSM amounts comparable to RB membranes. Altogether, these results show that the TFSM derivatives are valuable tools for studying the sphingolipid distribution and metabolism in infections, even though confirming the activity of SMases within chlamydial inclusions would require further functional analysis.

The distinct chemical structures of the two TFSM derivatives allow visualization of their downstream metabolites. When the ceramide

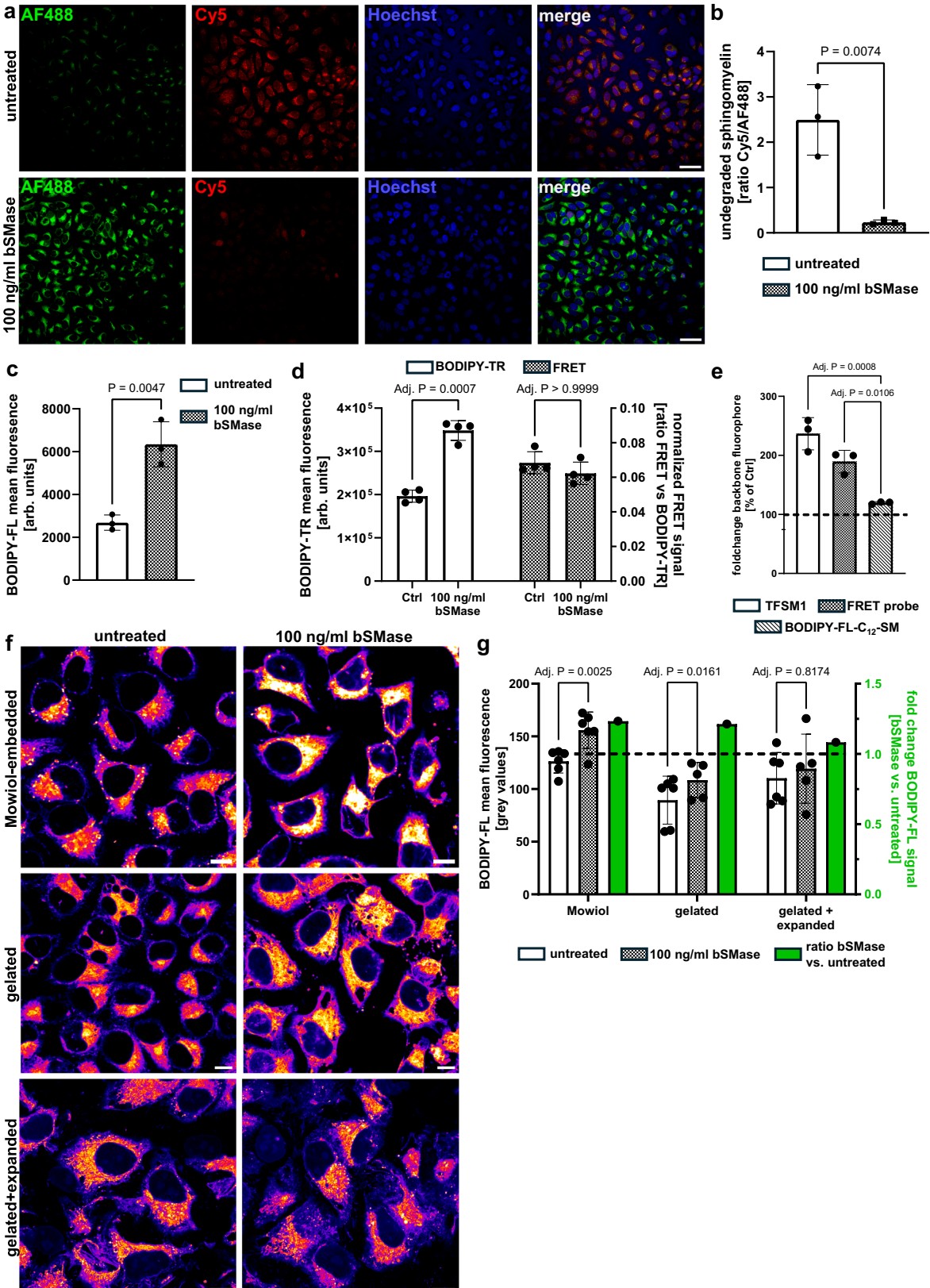

derivatives of TFSMs are cleaved into sphingosine and fatty acid, the fatty acid of **TFSM 2** will carry the fluorescent reporter, while for **TFSM 1** exclusively, the sphingoid backbone is visualized. This is advantageous because sphingosine is well retained in the samples upon fixation due to its natural amino group. We here detected a high proportion of metabolized TFSM molecules in chlamydial inclusions,

especially in EBs, regardless of the use of **TFSM 1** or **TFSM 2**, and thus we assume that this metabolite is ceramide.

Since conventional CLSM is restricted to a spatial resolution of ~200–250 nm[59], the investigation of small objects such as EBs with a diameter of ~200–300 nm[60] or lysosomes with a diameter of 200–600 nm[61] is challenging. Hence, high-resolution microscopy

**Fig. 7 | Bacterial sphingomyelinase (bSMase) treatment enhances signal of backbone fluorophores. a, b** bSMase affects backbone fluorophores FRET-independently. HeLa cells were incubated with **TFSM 1** in the presence of 1% FBS. The molecule was removed, and cells were bSMase-treated, fixed, and stained with Cy5-azide and AlexaFluor™ (AF)488. Fluorescence intensities in Cy5 and AF488 channels were measured, and Cy5/AF488 ratios were calculated. Scale bars: 50 μm. $n = 3$. **c** Quantification of backbone fluorescence by flow cytometry. HeLa cells were incubated with **TFSM 1** in the presence of 1% FBS. **TFSM 1** was removed, and cells were bSMase-treated, detached, fixed, BODIPY-FL-DBCO-stained, and analyzed by flow cytometry. $n = 3$. **d** bSMase treatment enhances the backbone fluorophore of a FRET probe. HeLa cells were incubated with the FRET probe in the presence of 1% FBS. After probe removal, cells were bSMase-treated, detached, and analyzed by flow cytometry for FRET and BODIPY-TR (backbone) fluorescence. Mean BODIPY-TR fluorescence (left $y$-axis) and FRET/BODIPY-TR ratios (right $x$-axis). $n = 4$. **e** Comparison of bSMase treatment effects on backbone fluorophores in different SM derivatives. HeLa cells were incubated with **TFSM 1**, the FRET probe, or BODIPY-

FL-$C_{12}$-SM. Molecules were removed, and cells were bSMase-treated and detached. Cells incubated with **TFSM 1** were fixed stained with BODIPY-FL-DBCO, and backbone fluorophores were analyzed by flow cytometry (BODIPY-FL for **TFSM 1**; BODIPY-FL-$C_{12}$-SM and BODIPY-TR for the FRET probe). Mean fluorescence in bSMase-treated samples was normalized to untreated controls (100%). $n = 3$. **f, g** bSMase-induced changes in backbone fluorophores are absent in expanded samples. Samples were incubated with **TFSM 1** for 24 h. Then, cells were washed, treated with bSMase, and either embedded in Mowiol or in the hydrogel, which was left unexpanded or 4x expanded. Scale bars: 40 μm. The mean fluorescence was measured within cell-containing areas (**g**), and fold changes were calculated (green bars, right $y$-axis). $n = 5$ for gelated + 100 ng/ml bSMase and gelated + expanded + 100 ng/ml bSMase, otherwise $n = 6$. Statistics: Two-sided unpaired Student's $t$ test (**b, c**). Two-way ANOVA and Šídák's multiple comparisons test (**d**), two-sided one-sample $t$ test (**e**), mixed effects analysis, and Šídák's multiple comparisons test (**f, g**). Bars represent means ± SD. n corresponds to biological replicates. Source data and detailed statistics are provided as a Source Data file.

techniques are required. Other approaches that analyze sphingolipid metabolism were restricted to single-cell level, even though they require specialized equipment such as two-photon microscopy[32] or matrix-assisted laser desorption/ionization mass spectrometry imaging (MALDI-MSI)[62]. The methods presented in this study enable visualization of SM turnover on a subcellular level with a spatial resolution of ~ 60 nm. Moreover, ExM does not require special equipment, and hence, the presented techniques are applicable to most research laboratories. We are convinced that the probes will find versatile applications in the future to visualize other processes that involve SMase activity. In addition, the TFSM analogs might serve as examples for the development of other functionalized sphingolipid derivatives to determine further metabolic processes, such as ceramidase activity.

Interestingly, we observed FRET-independent effects on the fluorescence of the membrane-associated fluorophores in TFSMs, the visible range FRET probe, and BODIPY-FL-$C_{12}$-SM. The signal intensity of these fluorophores was enhanced if we treated the cells with bSMase. These changes are probably not dependent on the cleavage of the molecules themselves but rather are due to the membrane environment of the fluorophores since the visible-range FRET probe, which cannot be metabolized by the enzyme, behaved similarly to TFSM and BODIPY-FL-$C_{12}$-SM, which both are accepted as substrate by the enzyme[63]. Moreover, amplification of membrane-associated fluorophores upon bSMase treatment was exclusively detected in specimens imaged by conventional CLSM but not in 4-fold expanded samples. We hypothesize that the measured fluorescence intensity depends on interfluorophore distance, which is increased by the expansion process. Hence, self-quenching is reduced in expanded samples. The effect observed upon bSMase treatment can be explained by the lipid raft hypothesis, which postulates that SM is not uniformly distributed within the membrane but accumulates at specific sites, so-called raft domains[64,65]. Consequently, neighboring SM molecules are in close proximity within raft domains and hence, membrane-associated fluorophores might undergo self-quenching. Previously, self-quenching was detected within lipid bilayers in vitro upon concentrating fluorescently-tagged lipids at specific membrane sites by an electric field[66]. Metabolization of SM by SMases is known to influence lipid raft composition[67], which could result in a larger spatial distance between individual SM molecules and hence, would disable self-quenching and yield higher fluorescence signals.

## Methods
### Ethical regulations and study protocol
Experiments involving the human pathogen *Chlamydia trachomatis* were conducted in the Biosafety Level (BSL) 2 laboratory of the Chair of Microbiology at the University of Wuerzburg, which is registered with the Government of Lower Franconia under code 8791-1.30.

### In vitro SMase activity assay
To determine ASM as well as the bSMase β-toxin activity towards a visible range FRET probe[31], the probe was incubated with 0.25 mU/ml recombinant human ASM (rhASM, R&D Cat. No. 5348-PD) or recombinant β-toxin (Sigma, Cat. No. S8633) in a volume of 200 μL/well in a 96 well plate (Corning, black/clear flat bottom; Ref. 3603) at 37 °C under orbital shaking in a Tecan Mplex microplate reader. The reaction was carried out in acidic buffer [200 mM sodium acetate pH 5 (Roth, Cat. No. 6773.2), 500 mM NaCl (VWR, Cat. No. 27810.364), 500 μM ZnCl$_2$ (Sigma, Cat. No. 208086) 0.2 % Nonidet P-40 (AppliChem, Cat. No. A1694.0250)] for ASM and neutral buffer [200 mM HEPES pH 7 (Roth, Cat. No. 6763.3), 0.05% Nonidet P-40] for β-toxin. Buffer composition was taken from a previous study[68]. FITC (ex. 488 ± 9 nm, em. 520 ± 20 nm) and BODIPY-TR (ex. 550 ± 9 nm, em. 620 ± 20 nm) was measured every 20 min. The ratio FITC vs. BODIPY-TR was calculated to determine the conversion of the FRET probe.

### Chemical synthesis
The TFSM derivatives were prepared in a multi-step synthesis starting from L-serine. Experimental details as well as characterization of the target molecules and intermediates by HRMS and NMR can be found in the Supplementary Information.

### Determining metabolization of TFSMs by LC-MS/MS
A detailed description can be found in the Supplementary Information.

### Cell culture
HeLa cells (ATCC CCL-2™) were cultured in RPMI+GlutaMAX™ medium (Gibco™, Cat. No. 72400054) containing 10% (v/v) heat-inactivated (56 °C at 30 min) fetal bovine serum (FBS, Sigma Aldrich. Cat. No. F7524) and 1 mM sodium pyruvate (Gibco™, Cat. No. 11360088).

HeLa229 cells (ATCC CCL-2.1™) were cultured in RPMI + GlutaMAX™ medium containing 10% (v/v) heat-inactivated FBS. Hela229 cells were confirmed to be free from Mycoplasma contamination using PCR. As the HeLa229 cell line was authenticated by ATCC, it was not further validated by our laboratory.

Human umbilical vein endothelial cells (HuVEC, Gibco™, Cat. No.C01510C) were cultured in MCDB131 medium (Gibco™, Cat. No. 10372019) complemented with microvascular growth supplement (Gibco™, Cat. No. S00525), 2 mM GlutaMAX™ (Gibco™, Cat. No. 35050061), 5 % (v/v) heat-inactivated (56 °C at 30 min) FBS, 2.76 μM hydrocortisone (Sigma Aldrich, Cat. No. H0888), 0.01 ng/ml human epidermal growth factor (Pep Rotech, Cat. No. AF-100-15) and 1x Penicillin-Streptomycin (Gibco™, Cat. No. 15140122).

Cells were seeded either one day (HeLa and HeLa229) or two days (HuVEC) prior to the experiment either in a 12-well plate ($1 \times 10^5$ cells per well) or 24-well plates ($0.5 \times 10^5$ cells per well). Standard tissue

culture procedures were used to maintain the cells. All cells were cultured in a humidified atmosphere with 5 % CO$_2$ (v/v) at 37 °C.

### *Chlamydia trachomatis* culture

The L$_2$/434/Bu (ATCC VR-902B) serovar of *Chlamydia trachomatis* was used for this study. *Chlamydia trachomatis* was expanded in HeLa229 cells at MOI 1 for 48 h. Cells were disrupted and mechanically lysed using glass beads (2.85–3.45 mm, Roth, Cat. No. A557.1). Cells were centrifuged for 10 min at 755 × *g* at 4 °C and supernatant was centrifuged for 30 min at 40,000 × *g* at 4 °C to collect bacteria. The pellet was washed once with 1 x sucrose–phosphate–glutamic acid (SPG) buffer [7.5% sucrose (Roth, Cat. No. 4621.2), 0.052% KH$_2$PO$_4$ (Roth, Cat. No. 3904.1), 0.122% Na$_2$HPO$_4$ (Roth, Cat. No. P030.2), 0.072% L-glutamine (Gibco, Cat. No. 25030081)]. To singularize the *Chlamydia*, the pellet was resuspended in 1 x SPG buffer and passed through G20 (B. Braun, Cat. No. 612-0141) and G18 (B. Braun, Cat. No. 612-0147) hollow needles. Aliquoted bacteria were stored at – 80 °C until further use. MOI 1 was determined by titration. *Chlamydia* was confirmed to be free from Mycoplasma contamination using PCR. The preparation of *Chlamydia* was based on a previously published protocol[69].

### Analysis of lipid derivatives by flow cytometry

HeLa cells were incubated with 10 μM **TFSM 1**, 10 μM visible range FRET probe, or 1 μM BODIPY™-FL-C$_{12}$-SM (Thermo Fisher, Cat. No. D7711) in treatment medium [MCDB131 medium containing 1% (v/v) heat-inactivated FBS and GlutaMAX™] for 2 h. Then, cells were washed thrice with DPBS (Gibco™, Cat. No. 14190169) and treated with 100 ng/ml β-toxin/bSMase for 3 h.

Consecutively, cells were washed thrice with DPBS, detached with 300 μL/well trypsin [TrypLE™, (Gibco™, Cat. No. 12604039)] for 5 min at 37 °C and resuspended in 300 μL 2% FBS in DPBS. For **TFSM 1**, samples were centrifuged (1.100 × *g*, 5 min, 4 °C) and fixed with 0.2% GA (Sigma, Cat. No. G5882) in 4% PFA in DPBS (Morphisto, Cat. No. 11762) for 30 min /RT. Cells were washed twice in 2% FBS in DPBS and then, permeabilized with 0.2% Triton X-100 (Roth, Cat. No. 6909) in PBS (AppliChem, Cat. No. A0964,9100). Next, samples were washed twice with 2% FBS in DPBS and stained with 2 μM BODIPY-FL-DBCO (Jena Biosciences, Cat. No. CLK-040-05) in 500 μL/sample Hanks' buffered saline (HBSS, Gibco™, Cat. No. 14025-100) for 1 h at 37 °C. Subsequently, cells were washed thrice with 2% FBS in DPBS. If not indicated otherwise, cells were centrifuged at 5.000 × *g*/5 min/4 °C between washing steps. Fixation, permeabilization, and staining were carried out in an end-over-end rotator. Then, cells were analyzed in an Attune NxT flow cytometer (Thermo Fischer) for BODIPY-FL (ex. 488 nm/em. bandpass 530/30 nm). The gating strategy is presented in Supplementary Fig. 10.

For samples treated with BODIPY-FL-C$_{12}$-SM and the visible range FRET probe, living cells were directly analyzed for BODIPY-FL/FITC (ex. 488 nm/em. bandpass 530/30 nm), FRET (ex. 488 nm/em, band pass 695/40 nm) and BODIPY-TR (ex.: 561 nm/em. bandpass 695/40 nm) with an Attune NxT flow cytometer.

### Analysis of lipid derivatives by confocal laser scanning microscopy (CLSM)

HeLa cells were incubated with 10 μM **TFSM 1** or **TFSM 2** in treatment medium for 2 h (unexpanded samples) or 24 h (ExM). If indicated, cells were washed thrice with DPBS and treated with 100 ng/ml bSMase for 3 h (unexpanded samples) or 2.5 μg/ml bSMase for 24 h (ExM). After treatment, samples were washed thrice with DPBS and fixed with 0.2% GA (Sigma, Cat. No. 10333)/4% PFA in DPBS for 30 min RT. Then, cells were washed thrice and permeabilized with 0.2 % Triton X-100. Subsequently, the samples were washed thrice with DPBS and stained with 2 μM (unexpanded) or 4 μM (ExM) BODIPY-FL-DBCO or AlexaFluor™ 488-DBCO (Jena Biosciences, Cat. No. CLK-1278) in HBSS for 1 h/37 °C. Cells were washed five times with DPBS and stained with 10 μM

(unexpanded) or 20 μM (ExM) AlexaFluor™546-azide (Jena Biosiences, Cat. No. CLK-1283), 20 μM Atto647N-azide (Atto-Tec, Cat.No. AD647N) or 2 μM Cy5-azide[70] in click reaction buffer [50 μM CuSO4, 2.5 mM sodium ascorbate (Sigma, Cat.No. A4034), 250 μM Tris(3-hydroxypropyltriazolylmethyl)amine (THPTA; Sigma, Cat. No. 762342) in PBS] for 1 h/37 °C. Then, samples were either mounted in Mowiol [24 g glycerol (Roth, Cat. No. 3783.2), 9.6 g Mowiol® 4-88 (Roth, Cat. No. 0713.2), 48 ml 0.2 M TRIS-HCl pH 8.5 (Sigma T1503), 24 ml Millipore H$_2$O] or used for ExM. Imaging was performed with a TCS SP5 confocal laser scanning microscope (Leica, Wetzlar, Germany). Images were adjusted for brightness and contrast. A median filter was applied to the images.

For determining the metabolic state, gray values in the FRET channel were multiplied by factor 2 (LAMP1-stained samples) or 3 (*Chlamydia*-infected samples) and then divided by the donor (BODIPY-FL) channel. The resulting ratio images were color-coded with the "Fires" look-up table (LUT) in Fiji.

For visualization of mitochondria, images were recorded on a Zeiss LSM900 confocal microscope with Airyscan 2, operating in the super-resolution (SR) imaging mode. An apochromatic oil immersion objective (40x/NA 1.3, Zeiss) was used for all images. The ZEN2 blue software (Zeiss, version 3.5) was used to select the optimal filter settings and excitation wavelengths for the various fluorescent dyes used in the experiment, utilizing the integrated preset configurations provided by the software. All acquired images were processed in standard strength Airyscan mode ensuring consistent and optimal data analysis across the sample set. Images were adjusted for brightness and contrast. A median filter was applied to the images.

### FRET acceptor bleaching

FRET acceptor bleaching was conducted with a Leica TCS SP5 microscope and the built-in FRET AB wizard. Thereby, images in donor and acceptor channels were recorded. Then, a region of interest (ROI) was selected in which the acceptor fluorophore was bleached by recording 20 frames with a high laser intensity. Subsequently, donor and acceptor channels were recorded, and FRET efficiency was determined by comparing donor and acceptor signal pre- and post-bleaching according to Eq. 1.).

$$FRET_{eff.} = \left( \frac{(Donor_{post} - Donor_{pre})}{Donor_{post}} \right) \cdot 100 \qquad (1)$$

### Detecting bSMase activity with BODIPY-FL-C$_{12}$-SM

HeLa cells were incubated with 1 μM BODIPY-FL-C$_{12}$-SM in treatment medium for 2 h in a 12-well plate. BOIDPY-FL-C$_{12}$-SM was removed, cells were washed thrice, and further incubated with 100 ng/ml β-toxin/bSMase for 3 h or left untreated. Then, cells were washed thrice, detached with 300 μL/well TrypLE, resuspended with 300 μL/well 2% FBS in PBS, and stored on ice. Samples were centrifuged 2000 × *g*/5 min/4 °C, pellets were resuspended in 250 μL 2:1 MeOH (Roth, Cat. No. 8388.6)/CHCl$_3$ (Roth, Cat. No. 3313.2), thoroughly vortexed and again centrifuged 13.000 × *g*/3 min/RT. 50 μL of the lower organic phase containing the lipids were taken and evaporated in a SpeedVac 5301 concentrator (Eppendorf) at 45 °C. Samples were resuspended in 10 μl 2:1 MeOH/CHCl$_3$ and spotted in 2.5 μl steps onto a thin-layer chromatography (TLC) plate (Alugram, Xtra Sil G/UV254, 0.2 mm/silica gel 60; VWR, Cat. No. 552-1006). TLC was developed with 80:20 MeOH/CHCl$_3$ and scanned with a Typhoon RGB scanner (Amersham). Intensities of fluorescent SM and ceramide bands were evaluated in Fiji to determine the proportion of unmetabolized BOIDPY-FL-C$_{12}$-SM.

### Antibody staining

After TFSMs were stained with DBCO and azide dyes, samples were blocked for 1 h/RT with 10% FBS in DPBS. Then, cells were incubated

with 4 μg/ml anti-LAMP-1 antibody (SantaCruz, Cat. No sc-18821, clone H5G11, Lot. #H2118), 2.5 μg/ml anti-GM130 antibody (Becton Dickinson, Cat. No. 610823, Lot. 7163670), 1:50 anti-Prx3 antibody (OriGene, Cat. No. TA322472) or 4 μg/ml anti-chlamydial HSP60 antibody (SantaCruz, Cat. No. sc-57840, Lot. #K1220) in blocking buffer overnight at 4 °C or for 1 h at RT. Then, cells were washed thrice with DPBS and incubated with 20 μg/ml CF™568 (Sigma, Cat. No. SAB4600082, Lot. 16C1017) or 20 μg/ml AlexaFluor™405 (ThermoFisher, Cat. No. A48255, Lot. YA353687) secondary antibody in blocking buffer at 4 °C overnight or for 1 h at RT.

### Expansion microscopy of TFSMs
Stained samples were post-fixed with 0.2% GA in DPBS for 15 min/RT and washed thrice with DPBS. Then, the glass slide was turned upside down in a drop of 80 μl (Ø15 mm slide) or 56 μl (Ø12 mm slide) monomer solution [8.625% sodium acrylate (Sigma, Cat. No. 408220), 2.5% acrylamide (Sigma, Cat. No. A9926), 0.15% N,N'-methylene bisacrylamide (Sigma, Cat. No. 146072), 2 M NaCl (Sigma, Cat. No. S5886) in PBS] that was freshly complemented with 0.2% (w/v) ammonium persulfate (Sigma, A3678) and 2% (v/v) tetramethyl ethylene diamine. Polymerization was carried out for 90 min at RT, and gels then were digested with 8 U/ml proteinase K (Sigma, Cat. No. P4860) in digestion buffer [50 mM Tris pH 8.0, 1 mM EDTA (Sigma, Cat. No. ED2P), 0.5% Triton X-100 and 0.8 M guanidine HCl (Sigma, Cat. No. 50933)] for 30 min/RT. Consecutively, gels were expanded in excess of Millipore water, whereby water was exchanged every hour until maximum expansion was reached. The expanded gels were cut in pieces of 1-2 cm and transferred into a Lab-Tek™ II chamber (VWR, Cat. No. 734-2055), which was coated with 0.01% poly-L-lysine solution (Sigma, Cat. No. A-005-C). Gels were imaged with a Leica TCS SP5 microscope. This protocol was adapted from a previous study[35].

### Infection with *Chlamydia trachomatis* and treatment with lipid derivatives
HeLa229 cells were seeded a day prior to the experiment in a 12-well plate ($1 \times 10^5$ cells per well). The next day, the medium of the cells was exchanged for fresh RPMI + GlutaMAX™ medium containing 10% heat-inactivated FBS. The cells were infected with *C. trachomatis* at MOI 1. 3 h after infection, the medium was exchanged to RPMI + GlutaMAX™ medium containing 1% heat-inactivated FBS. 10 μM **TFSM 1** or **TFSM 2** was added for 24 h. The cells were fixed with 0.2% GA/4% PFA in DPBS for 15 min RT and further processed for confocal laser scanning microscopy as described above.

### FLIM-FRET measurements and analysis
Fluorescence lifetime measurements were performed on a MicroTime200 (PicoQuant, Berlin, Germany) time-resolved confocal fluorescence microscope setup equipped with a FLIMbee galvo scanner (PicoQuant, Berlin, Germany), an Olympus IX83 microscope including an oil-immersion objective (60×, NA 1.45; Olympus), two single-photon avalanche photodiodes (SPADs) (Excelitas Technologies, 75154 K3, 75154 L6), and a TimeHarp300 dual-channel board. A white-light laser (NKT Photonics, SuperK extreme) was used for pulsed excitation at 488 nm with a repetition rate of 19.5 MHz and was coupled into the MicroTime200 system via a glass fiber (NKT Photonics, SuperK FD PM, A502-010-110). For all measurements, a 100 μm pinhole was used. The emission light was split onto the SPADs using a 50:50 beamsplitter (PicoQuant). Two identical bandpass filters (ET525/50 M, Chroma) were installed in front of the SPADs to filter out after-glow effects as well as scattered and reflected light. Measurements were performed and analyzed with the SymPhoTime64 software (PicoQuant, Version 2.8). All measurements were performed with a pixel dwell time of 25 μs. All unexpanded samples were measured with an irradiation intensity of ~ 0.5 kW cm$^{-2}$ in T3 mode with 25 ps time resolution, whereas all measurements of expanded samples were measured with ~ 7 kW cm$^{-2}$

in T3 mode. For single-cell measurements of unexpanded samples, images of 250 pixels * 250 pixels (100 nm * px$^{-1}$) were taken with a frame frequency of 0.3 Hz, and five frames were averaged to get the final image. For the overview image of unexpanded samples, images of 1000 pixels * 1000 pixels (150 nm * px$^{-1}$) were measured with a frame frequency of 0.013 Hz, and two frames were averaged to get the final image. For all expanded samples, five frames were averaged, consisting of 500 pixels * 500 pixels (180 nm * px$^{-1}$) at a frame frequency of 0.1 Hz to get the final image.

For analyzing the fluorescence lifetime of all images, the decay parameters were determined by least-squares deconvolution, and their quality was judged by the reduced χ2 values and the randomness of the weighted residuals (χ2 of roughly 1)[71]. A multiexponential model was used to fit the decay (Eq. 2).

$$\tau_{av} = \tau_1 a_1 + \tau_1 a_1 \tag{2}$$

For visualizing FLIM images, we encoded the lifetime and intensity using a color map and the brightness, respectively. Isoluminescent and perceptual uniform color maps were used to prevent an overlap of intensity and fluorescence lifetime using the TrackNTrace Lifetime Edition[72,73].

### Statistics & reproducibility
The sample size was determined empirically based on similar experiments performed previously. No statistical method was used to predetermine the sample size. No data were excluded from the analyses. ROIs during microscopy were selected randomly. Otherwise, the experiments were not randomized. For recording and evaluation of the FLIM measurements, investigators were blinded. Otherwise, the Investigators were not blinded to allocation during experiments and outcome assessment. Statistical analysis was performed with Graph-Pad Prism (10.1.2). Detailed results from statistics are provided in the Source Data file. The sample size "n" indicated in Figure Legends refers to biological replicates.

### Reporting summary
Further information on research design is available in the Nature Portfolio Reporting Summary linked to this article.

## Data availability
All data generated or analyzed during this study are included in this published article and its supplementary information files. The mass spectrometry data generated in this study are provided in the Supplementary Information and the Source Data. Source data are provided in this paper.

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

## Acknowledgements

This work was supported by the Deutsche Forschungsgemeinschaft (DFG) within the research training group RTG2581 (M.S, T.R., M.F., B.K., J.S.), the Collaborative Research Center SFB 1583 (T.R., M.F.) and under code AR 376/22-1 (C.A.). M.R. was supported by funds from the Bavarian State Ministry of Science and the Arts and the University of Würzburg to the Graduate School of Life Sciences (GSLS), University of Würzburg. T.R. and M.S. received funding from the European Research Council (ERC) under the European Union's Horizon 2020 research and innovation program Grant No. 835102 (M.S.) and Grant No. ERC-2018-ADG/NCI-CAD (T.R.). The Leica TCS SP5 CLSM was funded by the DFG under project code 116162193. The authors thank Danush Taban for AiryScan-Imaging, Vera Kozjak-Pavlovic for supplying antibodies, and the group Christian Häring for providing the Typhoon scanner. Figure 1a and Supplementary Fig. 1 were created with BioRender.com, released under a Creative Commons Attribution-NonCommercial-NoDerivs 4.0 International license.

## Author contributions

M.R., L.K., J.S., and M.F. conceptualized the study. M.R., L.K., F.W., J.S., and M.F. wrote the manuscript. L.K., T.P., R.E., C.K., and C.A performed chemical synthesis. M.R., F.W., D.W., F.S., D.A.H., M.S., B.K., and T.R. performed and analyzed biological experiments. All authors contributed to the article and approved the submitted version.

## Funding

## Competing interests

The authors declare no competing interests.
