## [Peer Review File · Nature Communications]

Reviewers' Comments:

Reviewer #1:

Remarks to the Author:

The manuscript presented by Professor Seibel and colleagues introduces new trifunctional sphingomyelin (TFSM) derivatives for visualizing sphingomyelin distribution and sphingomyelinase (SMase) activity in infection processes, and could be applied to expansion microscopy for high spatial resolution. Sphingomyelin (SM) is the most abundant and a crucial molecule in sphingolipid metabolism. This investigation demonstrated the utility of TFSM derivatives in the study of sphingomyelin metabolism, facilitating the distinction of membrane variations during chlamydial development. Specifically, TFSM 1 and TFSM 2 acted as effective substrates for SMases, integrating seamlessly into cellular lipid extracts. The design of TFSM probes embodies versatility, enabling the conjugation of diverse fluorescent dyes tailored to the requirements of individual experiments, thereby enhancing experimental flexibility. Furthermore, the employment of Expansion Microscopy (ExM) within this study permits an intricate examination of sphingomyelin metabolism. ExM addresses the issue of fluorophore self-quenching, significantly amplifying signal intensity. The integration of TFSM derivatives and ExM in this research underscores their significance in advancing the understanding of sphingomyelin metabolism, marking a substantial advancement in the field of sphingolipid research. The reviewer would like the authors address the following concerns,

- (1) The manuscript demonstrates an innovative method for trifunctional sphingomyelin derivatives by click chemistry SPAAC and CuAAC for introducing two small fluorescence probes enabling FRET system to monitor lipid metabolites, or sphingomyelin turnover but lacks the comparison with existing methods for sphingomyelin turnover and metabolism study.
- (2) In the previous studies by the authors, ref 30~33, there are several demonstrations for the live imaging such as “Amplification of a FRET Probe by Lipid–Water Partition for the Detection of Acid Sphingomyelinase in Live Cells”; “A Novel Visible Range FRET Probe for Monitoring Acid Sphingomyelinase Activity in Living Cells”; it would be nice to include the demonstration for the live imaging or there is a limitation on this application.
- (3) There are several cell lines used in the manuscript such as HEK293T or HeLa, maybe more validation for the different cell types such as HL60.
- (4) In fig 2f, why does TFSM predominantly accumulate within the Golgi apparatus? Is there evidence of TFSM accumulation in any other membrane-bound organelles following a 24-hour incubation period? The authors are encouraged to provide imaging

data to clarify this phenomenon. The zoomin images for Golgi and TFMS co-localization is required. It would be difficult to make the conclusion based on fig 2f. or other membrane-rich organelles should be labeled and compared.

(5) Why are most of the experiments using TFMS1 instead of TFMS2? The authors do not explicitly mention the reason for this preference. One is in the backbone of long carbon chain, the other one is next to the NH-group, is there any FRET efficiency difference? or the SMase efficiency difference?

(6) In fig 4d, for the FLIM images, the color scheme should be ranged in 0~5 ns, otherwise it is difficult for the readers to observe the difference, actually in the other figures, the authors present in 0~5 ns color scheme, not sure this figure in 0~8 ns. Maybe the authors could check this.

(7) The reviewer thinks the good strength for this designed probe is for lipid ExM, probably the authors could elaborate a bit the limitation on ExM of lipid. Why it is not easy to perform the ExM to visualize lipid. I knew there are already several publications on this, maybe a bit background introduction in the current manuscript to highlight the designed probe is very promising for retaining lipid components after expansion process. And it will be great to show the probe without NH modification (the linker to the ExM polymer), and give a result in a lot less amount of lipid retained after ExM process. The readers will appreciate the tri-functional design system.

(8) In the last figure, the authors mention, "We tested if the metabolization of TFMSs can be detected by ratiometrically analyzing two fluorophores without using a FRET system," but at the end of the paragraph, they did not answer their question. They should either clarify their question or modify the content.

(9) another finding that the authors point out is that the fluorescence enhancement in fig 6f due to the environment of the dye molecules, this is also very interesting for the reviewer. Although the ExM didn't gain the enhancement due to the physical separation of the dye molecules or just because of the hydrophobic environment change (?) instead of the separation of dye molecules.

(10) it would be nice to make the figures with better contrast for the fluorescence images such as fig 2d or 6 a etc.

Minor concerns:

1. What is the membrane composition of reticulate bodies (RBs) and elementary bodies (EBs) in chlamydial inclusions? Maybe there should be some descriptions included.

2. How does the fluorescence lifetime of TFMS probes differ between chlamydial

inclusions and host cell membranes?

3. proper fixation of the compounds was attained by treatment with buffer containing 0.2% glutaraldehyde and 4% PFA (Supp. Fig. 1b), is this better than PFA only in terms of the fluorescence maintaining? Other experiments were performed based on this?

4. The name of neutral SMase should be unified in the figure and context.

5. Ref 36 and ref 49 are duplicated.

Overall, the reviewer likes the study the authors present, especially that the FRET performance could be preserved after ExM, this could open up a lot of applications for the related studies. And the demonstration for the chlamydia inclusion by this specialized probe is beautiful.

Reviewer #2:

Remarks to the Author:

In this manuscript, Kersting et al describe novel probes allowing to visualize the cellular distribution and metabolism of sphingomyelin by microscopy techniques (fluorescence microscopy, FRET, expansion microscopy) and flow cytometry. The authors convincingly show the potential of these novel reagents by examining the activity of a sphingomyelinase at the plasma membrane and, in particular, by dissecting sphingomyelin localization and metabolism in Chlamydia-infected cells. Overall, I think this a very solid work and a very interesting manuscript. I just have a few minor comments:

- The second sentence of the Introduction (lines 39-40) is awkward and should be reformulated; sterols are lipids and receptors can be proteins but also lipids and polysaccharides.

- About Figure 2f it is written that “We observed the accumulation of TFSM 1 and TFSM 2...”; however, in the image and in the legend only TFSM1 is shown/mentioned.

- In Figure 2d, the BODIPY-FR legend should be a white bar.

- Have the authors considered the possibility of moving the last section of the Results (“Membrane-integral fluorophores are affected by their lipid environment”) to earlier in the manuscript? Before the description of the infections with Chlamydia?

Reviewer #3:

Remarks to the Author:

I co-reviewed this manuscript with one of the reviewers who provided the listed reports. This is part of the Nature Communications initiative to facilitate training in peer review

and to provide appropriate recognition for Early Career Researchers who co-review manuscripts.

Reviewer #4:

Remarks to the Author:

The manuscript by Kersting et al describes new synthetic sphingomyelin derivatives with 3 labeling sites. Presented work is a continuation of the previously published contribution published in Nature Communications in 2020 (10.1038/s41467-020-19897-1), where the authors introduced a somehow similar sphingolipid ceramide with a single click-chemistry labeling site. The currently presented TFMS 1 and TFMS 2 compounds are the effect of considerable further development. The two clickable sites and a primary amino group that can be used for labeling during fixation greatly increase versatility of the compounds and allow for much more precise quantification of the presence of sphingolipid and its metabolites. The two clickable sites are commonly cleaved apart by sphingomyelinase, which allows for an elegant and exact determination of sphingolipid metabolism using FRET approach.

The new compounds are used to visualize lipid membranes, measure sphingomyelinase activity in HeLa cells. They are also used to characterize chlamydia infection and perform expansion microscopy experiments. All these experiments clearly demonstrate the great potential of the newly developed probes. I found the paper of high importance for the studies of sphingolipid metabolism and its disorders and as such I believe it should be published in Nature Communications.

Minor critical remarks:

1. The sentence at lines 335-337 leaves some doubts. It is not clearly stated whether the authors considered possibility of unspecific FRET. It would be better to state more clearly if the observed FRET signal was higher than it would be expected for a simple mixture of the donor and acceptor.
2. I would strongly recommend to use isoluminescent color maps when presenting FLIM results (Figs 4, 5, supp. 2, and supp. 3). The color maps used by the authors, although slightly better than classical rainbow maps, are still not perceived as uniformly bright. This means that certain hues appear brighter than the others. This interferes with judging the intensities on the FLIM images. Ideally the two parameters (fluorescence intensity and lifetime) should be fully reciprocal and not mixed together.
3. In equation i. (line 627) the multiplication sign is in fact a letter x, which is confusing.

REVIEWER COMMENTS

Reviewer #1 (Remarks to the Author):

The manuscript presented by Professor Seibel and colleagues introduces new trifunctional sphingomyelin (TFSM) derivatives for visualizing sphingomyelin distribution and sphingomyelinase (SMase) activity in infection processes, and could be applied to expansion microscopy for high spatial resolution. Sphingomyelin (SM) is the most abundant and a crucial molecule in sphingolipid metabolism. This investigation demonstrated the utility of TFSM derivatives in the study of sphingomyelin metabolism, facilitating the distinction of membrane variations during chlamydial development. Specifically, TFSM 1 and TFSM 2 acted as effective substrates for SMases, integrating seamlessly into cellular lipid extracts. The design of TFSM probes embodies versatility, enabling the conjugation of diverse fluorescent dyes tailored to the requirements of individual experiments, thereby enhancing experimental flexibility. Furthermore, the employment of Expansion Microscopy (ExM) within this study permits an intricate examination of sphingomyelin metabolism. ExM addresses the issue of fluorophore self-quenching, significantly amplifying signal intensity. The integration of TFSM derivatives and ExM in this research underscores their significance in advancing the understanding of sphingomyelin metabolism, marking a substantial advancement in the field of sphingolipid research. The reviewer would like the authors address the following concerns,

We thank the reviewer for carefully reading our manuscript and the suggestions for improvement.

(1) The manuscript demonstrates an innovative method for trifunctional sphingomyelin derivatives by click chemistry SPAAC and CuAAC for introducing two small fluorescence probes enabling FRET system to monitor lipid metabolites, or sphingomyelin turnover but lacks the comparison with existing methods for sphingomyelin turnover and metabolism study.

We agree with the reviewer that the comparison to conventional methods was lacking in the original manuscript. To address this issue, we now conducted experiments where we incubated cells with commercially available BODIPY-FL-C₁₂-SM and subsequently treated with the bacterial SMase β -toxin. Afterwards we extracted the lipids and determined the ratio between BODIPY-FL-C₁₂-Ceramide and BODIPY-FL-C₁₂-SM via thin layer chromatography (TLC). In the revised manuscript, these data can be found in **Fig.2i** and images of TLCs can be found in **Supp Fig. 1e**. The results of this experiment are described in lines 266-270.

The TLC measurements confirm our microscopic observations.

(2) In the previous studies by the authors, ref 30~33, there are several demonstrations for the live imaging such as “Amplification of a FRET Probe by Lipid–Water Partition for the Detection of Acid Sphingomyelinase in Live Cells”; “A Novel Visible Range FRET Probe for Monitoring Acid Sphingomyelinase Activity in Living Cells”; it would be nice to include the demonstration for the live imaging or there is a limitation on this application.

There is a limitation for live cell imaging since copper salts that are required for CuAAC are cytotoxic. This is a benefit of probes with pre-attached fluorophores. In the revised manuscript this is mentioned in the discussion (lines 497 and 498).

(3) There are several cell lines used in the manuscript such as HEK293T or HeLa, maybe more validation for the different cell types such as HL60.

We agree with the Reviewer and performed experiments with other cells, here primary human umbilical vein endothelial cells (HuVEC). We included FRET acceptor bleaching experiments (Fig 2i, see reply to Comment 1) as well as staining of mitochondria (Supp Figure 2).

(4) In fig 2f, why does TFSM predominantly accumulate within the Golgi apparatus? Is there evidence of TFSM accumulation in any other membrane-bound organelles following a 24-hour incubation period? The authors are encouraged to provide imaging data to clarify this phenomenon. The zoomin images for Golgi and TFSM co-localization is required. It would be difficult to make the conclusion based on fig 2f. or other membrane-rich organelles should be labeled and compared.

Our observation of a high TFSM fluorescence signal within the Golgi does not necessarily results from a specific accumulation, as high abundance of membranes within the Golgi as

well leads to higher TFSM fluorescence. Hence, we rephrased the according section to “enrichment of TFSM1 and TFSM2 fluorescence” (Line 246-247). We also increased the zoom of images showing colocalization of TFSMs with Golgi (Fif. 2f). Additionally, we stained mitochondria with an anti-Prx3 antibody, which colocalizes with TFSMs. This was done with unexpanded samples (Fig. 1g and Supp. Fig.2, see previous comment) or via expansion microscopy, which allowed resolution of the mitochondrial envelop (Supp. Fig. 3). We added respective text elements for unexpanded (line 249-252) and expanded samples (line 278 and 279).

(5) Why are most of the experiments using TFSM1 instead of TFSM2? The authors do not explicitly mention the reason for this preference. One is in the backbone of long carbon chain, the other one is next to the NH-group, is there any FRET efficiency difference? or the SMase efficiency difference?

We did not detect significant differences in FRET efficiency or cleavage by bSMase between TFSM1 and TFSM2. The reason for the preferential use of TFSM1 is the detection of downstream metabolites. TFSM1 has the azido function in the sphingoid backbone, which enables visualization of sphingosine (which naturally harbors a primary amino function), if the molecule is further metabolized by a ceramidase. As we are interested in sphingolipid metabolism, TFSM1 hence was a more suitable molecule. As recommended by the reviewer, we included another data set for visualization of chlamydial inclusions by TFSM2 and ExM (Supp. Fig. 5, line 372-373). Like for TFSM1, we detected higher proportion of the metabolized molecule in EBs compared to RBs. We conclude that ceramide and not another downstream metabolite accumulates within EBs, since then we would expect that the accumulation is

exclusively visible with only one of the TFSMs. We included an according paragraph in the Discussion to explain the problematic of detection of downstream metabolites (line 540-549).

(6) In fig 4d, for the FLIM images, the color scheme should be ranged in 0~5 ns, otherwise it is difficult for the readers to observe the difference, actually in the other figures, the authors present in 0~5 ns color scheme, not sure this figure in 0~8 ns. Maybe the authors could check this.

As recommended by the reviewer, we have adjusted the range of the color scheme in all figures.

(7) The reviewer thinks the good strength for this designed probe is for lipid ExM, probably the authors could elaborate a bit the limitation on ExM of lipid. Why it is not easy to perform the ExM to visualize lipid. I knew there are already several publications on this, maybe a bit background introduction in the current manuscript to highlight the designed probe is very promising for retaining lipid components after expansion process. And it will be great to show the probe without NH modification (the linker to the ExM polymer), and give a result in a lot less amount of lipid retained after ExM process. The readers will appreciate the tri-functional design system.

We previously described that the primary amino function is required for proper fixation of another ceramide derivative (DOI: <https://doi.org/10.1038/s41467-020-19897-1>). However,

we agree with the reviewer that the function of the amino group in retaining the new trifunctional molecules upon permeabilization should be highlighted better.

We therefore incubated HeLa cells with the commercially available BODIPY-FL-C₁₂-SM and measured BODIPY-FL fluorescence via flow cytometry. We compared the fluorescence signal in living cells, fixed cells as well as cells that were fixed and permeabilized. While fixation did not affect fluorescence signals compared to living cells, permeabilization drastically reduced retention of BODIPY-FL-C₁₂-SM. We compared BODIPY-FL-C₁₂-SM-treated samples with samples treated with TFSM1/2 that were fixed, permeabilized, and stained with BODIPY-FL-DBCO. TFSM1/2 was retained upon permeabilization. This additional data now can be found in Fig. 1, where we also included a comparison between PFA and GA/PFA fixation as suggested by the Reviewer in Comment 3, minor concerns). The results are described in lines 234-243 of the revised manuscript. The data set showing effects of FBS on cellular uptake of the visible-range FRET probe with pre-attached fluorophores was moved to Supp. Fig. 1b.

Moreover, we added a paragraph on the topic in the discussion, where we refer to the literature (line 501-508).

(8) In the last figure, the authors mention, "We tested if the metabolization of TFSMs can be detected by ratiometrically analyzing two fluorophores without using a FRET system," but at the end of the paragraph, they did not answer their question. They should either clarify their question or modify the content.

We added a clarifying sentence at the end of the paragraph (line 422).

(9) another finding that the authors point out is that the fluorescence enhancement in fig 6f due to the environment of the dye molecules, this is also very interesting for the reviewer. Although the ExM didn't gain the enhancement due to the physical separation of the dye molecules or just because of the hydrophobic environment change (?) instead of the separation of dye molecules.

We appreciate the suggestion of the reviewer. We observed the same phenomenon also in living cells (with the visible range FRET probe, Fig 6d), while samples incubated with TFSM1 were fixed, permeabilized and Mowiol-embedded. The hydrophobic environment of membrane-integral fluorophores in living cells should massively differ from Mowiol-embedded samples, since most natural lipids are lost during permeabilization. Nevertheless, we performed an additional experiment to further address this issue. Therefore, we

additionally analyzed samples that were embedded into a hydrogel, but which were not expanded. During gelation, the fluorescence environment again should change. However, we detected a similar increase in fluorescence of membrane-integral fluorophores upon bSMase treatment as observed for Mowiol-embedded samples. The new data can be found in Fig. 6f and g.

(10) it would be nice to make the figures with better contrast for the fluorescence images such as fig 2d or 6 a etc.

We thank the reviewer for the suggested. In these images, the fluorescence intensities between the conditions are compared. Hence, we would need to change the contrast in all panels the same way, which would result in oversaturation of pixels in some conditions and hence would bias the assessment by the reader.

Minor concerns:

1. What is the membrane composition of reticulate bodies (RBs) and elementary bodies (EBs) in chlamydial inclusions? Maybe there should be some descriptions included. Less is known about the membrane composition of RBs and EBs, since -to our knowledge- only one study addressed this question until now. This study did not measure the Ceramide content within EBs and RBs. Detection of lipids in the developmental forms was based on purification of RBs and EBs. We think that separation of RBs and EBs is challenging and hence the results of the study should be regarded critically due to cross contaminations. We included a section in the discussion, where we relate to this study.

2. How does the fluorescence lifetime of TFSM probes differ between chlamydial inclusions and host cell membranes?

The fluorescence lifetime in the chlamydial inclusions is higher than in the host cell membrane, due to metabolized TFSMs and FRET, respectively. This can be seen in Fig. 4d,e. However, there are also differences in the fluorescence lifetime within the inclusions, which can be seen in the FLIM images of expanded samples (Fig. 5 g). Labelled TFSM in EBs showed a longer fluorescence lifetime than in RBs, which indicates a differential metabolism of TFSM in the two developmental forms of *Chlamydia*. Fluorescent lifetime measurements are comparable to the FRET intensity measurements in Fig. 4 a, b and 5 a, b.

3. proper fixation of the compounds was attained by treatment with buffer containing 0.2% glutaraldehyde and 4% PFA (Supp. Fig. 1b), is this better than PFA only in terms of the fluorescence maintaining? Other experiments were performed based on this?

Fixation with PFA resulted in 3x lower retention of the molecules within the samples compared to GA/PFA fixation. We demonstrate this in additional experiments via flow cytometry. The new data can be found in Fig. 2e (see also the Reviewer's comment 7, above).

4. The name of neutral SMase should be unified in the figure and context.

We now refer to the enzyme as "bSMase" in all figures and the text.

5. Ref 36 and ref 49 are duplicated.

We thank the reviewer for attentively reading our manuscript and catching this error. The duplicate reference has been removed.

Overall, the reviewer likes the study the authors present, especially that the FRET performance could be preserved after ExM, this could open up a lot of applications for the related studies. And the demonstration for the chlamydia inclusion by this specialized probe is beautiful.

Reviewer #2 (Remarks to the Author):

In this manuscript, Kersting et al describe novel probes allowing to visualize the cellular distribution and metabolism of sphingomyelin by microscopy techniques (fluorescence microscopy, FRET, expansion microscopy) and flow cytometry. The authors convincingly show the potential of these novel reagents by examining the activity of a sphingomyelinase at the plasma membrane and, in particular, by dissecting sphingomyelin localization and metabolism in *Chlamydia*-infected cells. Overall, I think this a very solid work and a very interesting manuscript. I just have a few minor comments:

We thank the reviewer for the positive feedback and the interest in our work.

- The second sentence of the Introduction (lines 39-40) is awkward and should be reformulated; sterols are lipids and receptors can be proteins but also lipids and polysaccharides.

We reformulated the sentence and removed “sterols” and “receptors”.

- About Figure 2f it is written that “We observed the accumulation of TFSM 1 and TFSM 2...”; however, in the image and in the legend only TFSM1 is shown/mentioned.

We changed this paragraph and now solely refer to TFSM1 (line 245).

- In Figure 2d, the BODIPY-FR legend should be a white bar.

Since there is neither BODIPY-FL nor BODIPY-TR depicted in Figure 2d, we are unable to identify the issue the Reviewer is referring to.

- Have the authors considered the possibility of moving the last section of the Results (“Membrane-integral fluorophores are affected by their lipid environment”) to earlier in the manuscript? Before the description of the infections with Chlamydia?

As mentioned by the reviewer, the section “Membrane-integral fluorophores are affected by their lipid environment” was positioned earlier in the original draft of the manuscript. However, its current position at the end of the manuscripts makes it easier for the reader to understand the novel TFSM molecules by demonstrating proof-of-concept applications. Moreover, the Chlamydia experiments -in our opinion one of the highlights of our study- now are placed directly after introduction of the used molecules and ExM. We thus decided to move the section to the end of the manuscript.

Reviewer #3 (Remarks to the Author):

Reviewer #4 (Remarks to the Author):

The manuscript by Kersting et al describes new synthetic sphingomyelin derivatives with 3 labeling sites. Presented work is a continuation of the previously published contribution published in Nature Communications in 2020 (10.1038/s41467-020-19897-1), where the authors introduced a somehow similar sphingolipid ceramide with a single click-chemistry labeling site. The currently presented TFSM 1 and TFSM 2 compounds are the effect of considerable further development. The two clickable sites and a primary amino group that can be used for labeling during fixation greatly increase versatility of the compounds and allow for much more precise quantification of the presence of sphingolipid and its

metabolites. The two clickable sites are commonly cleaved apart by sphingomyelinase, which allows for an elegant and exact determination of sphingolipid metabolism using FRET approach.

The new compounds are used to visualize lipid membranes, measure sphingomyelinase activity in HeLa cells. They are also used to characterize chlamydia infection and perform expansion microscopy experiments. All these experiments clearly demonstrate the great potential of the newly developed probes. I found the paper of high importance for the studies of sphingolipid metabolism and its disorders and as such I believe it should be published in Nature Communications.

We thank the reviewer for carefully reading our manuscript and the positive evaluation.

Minor critical remarks:

1. The sentence at lines 335-337 leaves some doubts. It is not clearly stated whether the authors considered possibility of unspecific FRET. It would be better to state more clearly if the observed FRET signal was higher than it would be expected for a simple mixture of the donor and acceptor.

The signal within RBs/inclusions can clearly be distinguished from unspecific FRET, since we detected FRET via i) FLIM and ii) acceptor photobleaching. Compared to RBs, FLIM showed higher BODIPY-FL lifetimes in EBs, which means that there is only a small proportion of molecules that undergo FRET within RBs. Conversely, RBs own a higher FRET signal than EBs and, hence, we excluded that FRET signal in RBs results from unspecific background. We included a sentence to clarify this situation to the reader (line 364-365).

2. I would strongly recommend to use isoluminescent color maps when presenting FLIM results (Figs 4, 5, supp. 2, and supp. 3). The color maps used by the authors, although slightly better than classical rainbow maps, are still not perceived as uniformly bright. This means that certain hues appear brighter than the others. This interferes with judging the intensities on the FLIM images. Ideally the two parameters (fluorescence intensity and lifetime) should be fully reciprocal and not mixed together.

We agree with the Reviewer that the use of isoluminescent color maps is recommended and more suitable for the representation of FLIM data. Accordingly, we have adapted all color scales accordingly and now use a perceptual uniform color map to prevent non-uniform distances between hues. Furthermore, the color scale used is now also suitable for readers with color vision impairment.

3. In equation i. (line 627) the multiplication sign is in fact a letter x, which is confusing.

The letter "x" was replaced by a dot operator.

General comments by the authors:

We added the description of all materials and methods required in the revision process in the Materials & Methods section. We thereby also included an explanation of the gating strategy for flow cytometry analysis (**Supp. Fig. 7**). The acknowledgements and the author list were

adjusted as well. The required form for author list changes will be handed in as soon as possible.

Reviewers' Comments:

Reviewer #1:

Remarks to the Author:

Dear Authors,

the reviewer is satisfied with the revision. thanks for the sharing.

Bi-Chang

Reviewer #2:

Remarks to the Author:

The authors answered adequately to all my remarks of the first revisions, and I have no additional comments.

Reviewer #3:

Remarks to the Author:

Reviewer #4:

Remarks to the Author:

I appreciate all the changes and explanations provided by the authors.